# ADDRESSING LABEL SHIFT IN DISTRIBUTED LEARNING VIA ENTROPY REGULARIZATION

**Zhiyuan Wu**[*]
University of Oslo
zhiyuanw@ifi.uio.no

**Changkyu Choi**[*]
UiT The Arctic University of Norway
changkyu.choi@uit.no

**Xiangcheng Cao**
EPFL
xiangcheng.cao.epfl@gmail.com

**Volkan Cevher**
LIONS, EPFL
volkan.cevher@epfl.ch

**Ali Ramezani-Kebrya**
Department of Informatics, University of Oslo
Norwegian Centre for Knowledge-driven Machine Learning (Integreat)
ali@uio.no

## ABSTRACT

We address the challenge of minimizing *true risk* in multi-node distributed learning.[1] These systems are frequently exposed to both inter-node and intra-node *label shifts*, which present a critical obstacle to effectively optimizing model performance while ensuring that data remains confined to each node. To tackle this, we propose the Versatile Robust Label Shift (VRLS) method, which enhances the maximum likelihood estimation of the test-to-train label importance ratio. VRLS incorporates Shannon entropy-based regularization and adjusts the importance ratio during training to better handle label shifts at the test time. In multi-node learning environments, VRLS further extends its capabilities by learning and adapting importance ratios across nodes, effectively mitigating label shifts and improving overall model performance. Experiments conducted on MNIST, Fashion MNIST, and CIFAR-10 demonstrate the effectiveness of VRLS, outperforming baselines by up to $20\%$ in imbalanced settings. These results highlight the significant improvements VRLS offers in addressing label shifts. Our theoretical analysis further supports this by establishing high-probability bounds on estimation errors. The code is available at https://github.com/zhiyuan-11/VRLS_main/tree/main.

## 1 INTRODUCTION

The classical learning theory relies on the assumption that data samples, during training and testing, are *independently and identically distributed (i.i.d.)* drawn from an unknown distribution. However, this *i.i.d.* assumption is often overly idealistic in real-world settings, where the distributions of training and testing samples can differ significantly and change dynamically as the operational environment evolves. In distributed learning (Kim et al., 2022; Wen et al., 2023; Ye et al., 2023; Luo et al., 2023), where nodes retain their own data without sharing, these discrepancies across nodes become more pronounced, further intensifying the learning challenge (Rahman et al., 2023; Wang et al., 2023).

*Label shifts* (Lipton et al., 2018; Garg et al., 2022; Mani et al., 2022; Zhou et al., 2023) represent a form of distributional discrepancy that arises when the marginal distribution of labels in the training set differs from that in the test set, i.e., $p^{\text{te}}(\boldsymbol{y}) \neq p^{\text{tr}}(\boldsymbol{y})$, while the conditional distribution of features given labels, $p(\boldsymbol{x}|\boldsymbol{y})$, remains largely stable across both datasets. Label shifts commonly manifest both *inter-node* and *intra-node*, complicating the learning process in real-world distributed learning scenarios. However, a commonly used learning principle in this distributed setting, empirical risk minimization

---

[*]These authors contributed equally to this work.
[1]We use the term node to refer to a client, FPGA, APU, CPU, GPU, or worker.

(ERM) (Kur et al., 2024), operates under the assumption that the training and test distributions are identical on each node and across nodes. This overlooks these shifts, failing to account for the statistical heterogeneity across decentralized data sources. While the current literature (Yin et al., 2024) addresses statistical heterogeneity across nodes, it often neglects distribution shifts at test or operation time, which has been a significant challenge in the entire data science over decades.

The primary technical challenge in addressing label shifts lies in the efficient and accurate estimation of the test-to-train importance ratios, expressed as $p^{\text{te}}(\boldsymbol{y})/p^{\text{tr}}(\boldsymbol{y})$ for all labels. A widely popular solution is Maximum Likelihood Label Shift Estimation (MLLS) (Garg et al., 2020), which frames this estimation as a convex optimization problem, akin to the Expectation-Maximization (EM) algorithm (Saerens et al., 2002). Model calibration refers to the process of ensuring that predicted probabilities reflect the true likelihood of correctness, which is crucial for improving the accuracy of importance ratio estimation (Guo et al., 2017a; Garg et al., 2020). Bias-Corrected Calibration (BCT) (Alexandari et al., 2020) serves as an efficient calibration method that enhances the EM algorithm within MLLS.

While BCT and other post-hoc calibration techniques (Guo et al., 2017c; Kull et al., 2019; Wang et al., 2021; Sun et al., 2024) contribute to improved calibration and may potentially improve model performance, their primary focus remains on refining classification outcomes rather than on accurately approximating the true conditional distribution $p^{\text{tr}}(\boldsymbol{y}|\boldsymbol{x})$. The "predictor" in these literature captures the relationship between the input features $\boldsymbol{x}$ and the corresponding output probabilities across the labels in the discrete label space $\mathcal{Y}$, with $|\mathcal{Y}| = m$, which should approximate the true distribution of $p^{\text{tr}}(\boldsymbol{y}|\boldsymbol{x})$. Despite this goal, training with conventional cross-entropy loss often leads to models that produce predictions that are either highly over-confident or under-confident, resulting in poorly calibrated outputs (Guo et al., 2017a). Consequently, the predictor fails to capture the underlying uncertainty inherent in $p^{\text{tr}}(\boldsymbol{y}|\boldsymbol{x})$, which limits its effectiveness in estimating importance ratios (Alexandari et al., 2020; Garg et al., 2020; Guo et al., 2020; 2017b; Pereyra et al., 2017; McMahan et al., 2017).

To address this limitation, we propose a novel Versatile Robust Label Shift (VRLS) method, specifically designed to improve importance ratio estimation for tackling the label shift problem. A key idea of our VRLS method is to approximate $p^{\text{tr}}(\boldsymbol{y}|\boldsymbol{x})$ in a way that accounts for the inherent uncertainty over the label space $\mathcal{Y}$ for each input $\boldsymbol{x}$. Accordingly, we propose a new objective function incorporating regularization to penalize predictions that lack proper uncertainty calibration. We show that training the predictor in this manner significantly reduces estimation error under various label shift conditions.

Building upon our VRLS method, we extend its application to multi-node settings by proposing an Importance Weighted-ERM (IW-ERM) framework (Fang et al., 2023). Within the multi-node distributed environment, our IW-ERM aims to find an unbiased estimate of the overall true risk minimizer across multiple nodes with varying label distributions. By effectively addressing both intra-node and inter-node label shifts with generalization guarantees, our framework handles the statistical heterogeneity inherent in decentralized data sources. Our extensive experiments demonstrate that the IW-ERM framework, which trains predictors exclusively on local node data, significantly improves overall test error. Moreover, it maintains convergence rates and privacy levels comparable to standard ERM methods while achieving minimal communication and computational overhead compared to existing baselines. Our main contributions are as follows:

- We propose VRLS, which enhances the approximation of the probability distribution $p^{\text{tr}}(\boldsymbol{y}|\boldsymbol{x})$ by incorporating a novel regularization term based on Shannon entropy (Neo et al., 2024). This regularization leads to more accurate estimation of the test-to-train label importance ratio, resulting in improved predictive performance under various label shift conditions.

- By integrating our VRLS ratio estimation into multi-node distributed learning environment, we achieve performance close to an upper bound that uses true ratios on Fashion MNIST and CIFAR-10 datasets with 5, 100, and 200 nodes. Our IW-ERM framework effectively manages both inter-node and intra-node label shifts while remaining data confined within each node, resulting in up to 20% improvements in average test error over current baselines.

- We establish high-probability estimation error bounds for VRLS, as well as high-probability convergence bounds for IW-ERM with VRLS in nonconvex optimization settings (Section 5, Appendices E,

Table 1: Details of the label shift scenarios. Their IW-ERM formulas are presented in Appendix C.

| Scenario | #Nodes | Assumptions on Distributions | Ratio Node i Needs |
|---|---|---|---|
| `No-LS` in Equation (17) | 2 | $p_1^{\text{tr}}(\boldsymbol{y}) = p_1^{\text{te}}(\boldsymbol{y}), p_1^{\text{tr}}(\boldsymbol{y}) \neq p_2^{\text{tr}}(\boldsymbol{y})$ | $p_1^{\text{tr}}(\boldsymbol{y})/p_2^{\text{tr}}(\boldsymbol{y})$ |
| `LS on single` in Equation (18) | 2 | $p_1^{\text{tr}}(\boldsymbol{y}) \neq p_1^{\text{te}}(\boldsymbol{y}), p_2^{\text{tr}}(\boldsymbol{y}) = p_2^{\text{te}}(\boldsymbol{y})$ | $p_1^{\text{te}}(\boldsymbol{y})/p_1^{\text{tr}}(\boldsymbol{y}), p_1^{\text{te}}(\boldsymbol{y})/p_2^{\text{tr}}(\boldsymbol{y})$ |
| `LS on both` in Equation (18) | 2 | $p_1^{\text{tr}}(\boldsymbol{y}) \neq p_1^{\text{te}}(\boldsymbol{y}), p_2^{\text{tr}}(\boldsymbol{y}) \neq p_2^{\text{te}}(\boldsymbol{y})$ | $p_1^{\text{te}}(\boldsymbol{y})/p_1^{\text{tr}}(\boldsymbol{y}), p_1^{\text{te}}(\boldsymbol{y})/p_2^{\text{tr}}(\boldsymbol{y})$ |
| `LS on multi` in Equation (19) | $K$ | $p_k^{\text{tr}}(\boldsymbol{y}) \neq p_1^{\text{te}}(\boldsymbol{y})$ for all $k$ | $p_1^{\text{te}}(\boldsymbol{y})/p_k^{\text{tr}}(\boldsymbol{y})$ for all $k$ |

F). Additionally, we demonstrate that incorporating importance weighting does not negatively impact convergence rates or communication guarantees across various optimization settings.

## 2 IMPORTANCE RATIO ESTIMATION AND IMPORTANCE WEIGHTED-ERM

**Importance ratio estimation** Importance ratio estimation for label shifts has been addressed by methods such as solving linear systems (Lipton et al., 2018; Azizzadenesheli et al., 2019) and minimizing distribution divergences (Garg et al., 2020), primarily in the context of a single node. Lipton et al. (2018); Azizzadenesheli et al. (2019); Garg et al. (2020) assumed the conditional distribution $p(\boldsymbol{x}|\boldsymbol{y})$ remains fixed between the training and test datasets, while the label distribution $p(\boldsymbol{y})$ changes. Black Box Shift Estimation (BBSE) (Lipton et al., 2018; Rabanser et al., 2019) and Regularized Learning under Label Shift (RLLS) (Azizzadenesheli et al., 2019) are confusion matrix-based methods for estimating importance ratios in label shift problems. While BBSE has been shown consistent even when the predictor is not calibrated, its subpar performance is attributed to information loss inherent in using confusion matrices (Garg et al., 2020). To overcome this, Garg et al. (2020) has introduced the MLLS, resulting in significant improvements in estimation performance, especially when combined with post-hoc calibration methods like BCT (Shrikumar et al., 2019). This EM algorithm based MLLS method (Saerens et al., 2002; Garg et al., 2020) is concave and can be solved efficiently.

**Importance Weighted-ERM** Classical ERM seeks to minimize the expected loss over the training distribution using finite samples. However, when there is a distribution shift between the training and test data, the objective of ERM is not to minimize the expected loss over the test distribution, regardless of the number of training samples. To address this, IW-ERM is developed (Shimodaira, 2000; Sugiyama et al., 2006; Byrd & C. Lipton, 2019; Fang et al., 2020), which adjusts the training loss by weighting samples according to the importance ratio, i.e., the ratio of test-to-train density. Shimodaira (2000) has shown that the IW-ERM estimator is asymptotically unbiased under certain conditions. Building on this, Ramezani-Kebrya et al. (2023b) have recently introduced Federated IW-ERM, which incorporates importance ratio estimation to handle covariate shifts in distributed learning. However, this approach has limitations, as it does not address label shifts and the importance ratio estimation method poses potential privacy risks.

In this work, we focus on label shifts and propose an IW-ERM framework enhanced by our VRLS method. We show that our IW-ERM with VRLS performs comparably to an upper bound that utilizes true importance ratios, all while preserving data privacy across distributed data sources. This approach effectively addresses both intra-node and inter-node label shifts while ensuring convergence in probability to the overall true risk minimizer.

## 3 VERSATILE ROBUST LABEL SHIFT: REGULARIZED RATIO ESTIMATION

In this section, we introduce the Versatile Robust Label Shift (VRLS) method for importance ratio estimation in a single-node setting, which forms the basis of the IW-ERM framework. To solve the optimization problem of IW-ERM, each node $k$ requires an accurate estimate of the ratio:

$$r_k(\boldsymbol{y}) = \frac{\sum_{j=1}^{K} p_j^{\text{te}}(\boldsymbol{y})}{p_k^{\text{tr}}(\boldsymbol{y})}, \tag{1}$$

where $p_j^{\text{te}}(\boldsymbol{y})$ and $p_k^{\text{tr}}(\boldsymbol{y})$ represent the test and training label densities, respectively. To improve clarity and avoid over-complicating notations, we first consider the scenario where we have only one

---

**Algorithm 1** VRLS Importance Ratio Estimation Algorithm

---

**Require:** Labeled training data $\{(\boldsymbol{x}_i, \boldsymbol{y}_i)\}_{i=1}^{n^{\text{tr}}}$.
**Require:** Unlabeled test data $\{\boldsymbol{x}_j\}_{j=1}^{n^{\text{te}}}$.
**Require:** Initial predictor $f_{\boldsymbol{\theta}}$.
**Ensure:** Optimized predictor $f_{\boldsymbol{\theta}^*}$ and estimated importance ratio $\boldsymbol{r}_{f^*}$.
 1: **Training**:
 2:      Optimize $f_{\boldsymbol{\theta}}$ using Equation (4) via SGD.
 3:      Continue until the training loss drops below a threshold or the maximum epochs are reached.
 4:      Obtain the optimized predictor $f_{\boldsymbol{\theta}^*}$.
 5: **Importance Ratio Estimation**:
 6:      With the optimized predictor $f_{\boldsymbol{\theta}^*}$, estimate the importance ratio $\boldsymbol{r}_{f^*}$ using equation Equation (3).

---

node under label shifts and then extend to multiple nodes. Table 1 presents various scenarios. In a single-node label shift scenario, the goal is to estimate the ratio $r(\boldsymbol{y}) = p^{\text{te}}(\boldsymbol{y})/p^{\text{tr}}(\boldsymbol{y})$. Following the seminal work of Garg et al. (2020), we formulate importance ratio estimation as a Maximum Likelihood Estimation (MLE) problem by constructing an optimization problem based on Kullback-Leibler (KL) divergence to directly estimate $r(\boldsymbol{y})$. We train a predictor $f_{\boldsymbol{\theta}}$ to approximate $p^{\text{tr}}(\boldsymbol{y}|\boldsymbol{x})$, where $\boldsymbol{\theta}$ denotes the parameters of a neural network. After training, we apply the predictor $f_{\boldsymbol{\theta}^*}$ to a finite set of unlabeled samples drawn from the test distribution to obtain predicted label probabilities. These predictions are then used to estimate the ratio $\boldsymbol{r}_{f^*}$. Further details are provided in Algorithm 1.

One of the novelties of VRLS is its ability to better calibrate the predictor, enabling it to better approximate the true conditional distribution $p^{\text{tr}}(\boldsymbol{y}|\boldsymbol{x})$. This approximation faces two main challenges, as highlighted in Theorem 3 of (Garg et al., 2020): finite-sample error and miscalibration error. Entropy-based regularization can directly tackle miscalibration, which occurs when predicted probabilities systematically deviate from true likelihoods. Building on these insights, we introduce an *explicit* entropy regularizer into the training objective, which is based on Shannon's entropy (Pereyra et al., 2017; Neo et al., 2024). The regularization term $\Omega(f_{\boldsymbol{\theta}})$ is defined as:

$$\Omega(f_{\boldsymbol{\theta}}) = \sum_{c=1}^{m} \phi\big(f_{\boldsymbol{\theta}}(\boldsymbol{x})\big)_c \log\left(\phi\big(f_{\boldsymbol{\theta}}(\boldsymbol{x})\big)_c\right), \tag{2}$$

where $\phi$ denotes the softmax function, and $c$ represents the $c^{\text{th}}$ element of the softmax output vector. With this regularization to the softmax outputs, VRLS encourages smoother and more reliable predictions that account for inherent uncertainty in the data, leading to more accurate importance ratio estimates and improving the SotA in practice. These improvements are empirically demonstrated in Section 6. Our proposed VRLS objective is formulated as follows:

$$\boldsymbol{r}_{f^*} = \underset{\boldsymbol{r} \in \mathbb{R}_+^m}{\arg\max} \, \mathbb{E}_{\text{te}}\left[\log(f_{\boldsymbol{\theta}^*}(\boldsymbol{x})^\top \boldsymbol{r})\right], \tag{3}$$

where

$$\boldsymbol{\theta}^* = \underset{\boldsymbol{\theta}}{\arg\min} \, \mathbb{E}_{\text{tr}}\left[\ell_{CE}\big(f_{\boldsymbol{\theta}}(\boldsymbol{x}), \boldsymbol{y}\big) + \zeta\Omega(f_{\boldsymbol{\theta}})\right]. \tag{4}$$

The vector $\boldsymbol{r}$ in Equation (3), representing the importance ratios for all $m$ classes, belongs to the non-negative real space $\mathbb{R}_+^m$. This constraint set is defined similarly to MLLS (Garg et al., 2022), and we use the expected value $\mathbb{E}_{\text{te}}$ for estimation, denoting the optimal importance ratio as $\boldsymbol{r}_{f^*}$. To train the predictor $\boldsymbol{\theta}$, we minimize the cross-entropy loss $\ell_{CE}$ together with a scaled regularization term $\zeta\Omega(f_{\boldsymbol{\theta}})$, where $\zeta > 0$ is a coefficient controlling the regularization strength. Incorporating the regularizer $\Omega(f_{\boldsymbol{\theta}})$ improves the model calibration under the influence of $\ell_{CE}$ loss.

## 4   VRLS FOR MULTI-NODE ENVIRONMENT

We now extend VRLS to the multi-node environment, taking into account the privacy and communication requirements. This extension naturally aligns with the concept of IW-ERM, effectively integrating these considerations into the multi-node learning paradigm. We consider multiple nodes where each node has distinct training and test distributions. The goal here is to train a global model

---

**Algorithm 2** IW-ERM with VRLS in Distributed Learning

---

**Require:** Labeled training data $\{(\boldsymbol{x}_{k,i}^{\text{tr}}, \boldsymbol{y}_{k,i}^{\text{tr}})\}_{i=1}^{n_k^{\text{tr}}}$ at each node $k$, for $k = [K]$.

**Require:** Unlabeled test data $\{\boldsymbol{x}_{k,j}^{\text{te}}\}_{j=1}^{n_k^{\text{te}}}$ at each node $k$, for $k = [K]$.

**Require:** Initial global model $h_{\boldsymbol{w}}$.

**Ensure:** Trained global model $h_{\boldsymbol{w}}$ optimized with IW-ERM.
 1: **Phase 1: Importance Ratio Estimation with VRLS**
 2: **for each node** $k = 1$ to $K$ **in parallel do**
 3:     Train local predictor $f_{k,\theta}$ on local training data $\{(\boldsymbol{x}_{k,i}^{\text{tr}}, \boldsymbol{y}_{k,i}^{\text{tr}})\}$.
 4:     Use $f_{k,\theta^*}$ to estimate the importance ratio $\boldsymbol{r}_{k,f^*}$ on unlabeled test data $\{\boldsymbol{x}_k^{\text{te}}\}$ at node $k$.
 5: **end for**
 6: **Phase 2: Importance Ratio Aggregation**
 7: **for each node** $k = 1$ to $K$ **do**
 8:     Aggregate importance ratio using Equation (1).
 9: **end for**
10: **Phase 3: Global Model Training with IW-ERM**
11: Train global model $h_{\boldsymbol{w}}$ using Equation (IW-ERM) with the aggregated importance ratios.

---

that utilizes local data and addresses overall test error. In this setup, each node uses its local data to estimate the required importance ratios, as outlined in Section 3, and shares only low-dimensional ratio information, without the need to share any local data.

The process begins with each node training a global model on its local data, independently estimating its importance ratios. These locally computed ratios are then shared amongst the nodes, allowing for the aggregated ratio required for IW-ERM to be computed centrally. This aggregated ratio is then used to further refine the global model in a second round of global training. This approach ensures minimal communication overhead and preserves node data privacy, as detailed in Section 5. Our experimental results in Section 6 demonstrate that the IW-ERM framework significantly improves test error performance while minimizing communication and computation overhead compared to baseline ERM. The importance ratio estimation and IW-ERM are described in Algorithm 2.

To provide a more comprehensive understanding of the multi-node environment, the following discussion delves into its details. Let $\mathcal{X} \subseteq \mathbb{R}^{d_0}$ be a compact metric space for input features, $\mathcal{Y}$ be a discrete label space with $|\mathcal{Y}| = m$, and $K$ be the number of nodes in an multi-node setting.[2] Let $\mathcal{S}_k = \{(\boldsymbol{x}_{k,i}^{\text{tr}}, \boldsymbol{y}_{k,i}^{\text{tr}})\}_{i=1}^{n_k^{\text{tr}}}$ denote the training set of node $k$ with $n_k^{\text{tr}}$ samples drawn i.i.d. from a probability distribution $p_k^{\text{tr}}$ on $\mathcal{X} \times \mathcal{Y}$. The test data of node $k$ is drawn from another probability distribution $p_k^{\text{te}}$ on $\mathcal{X} \times \mathcal{Y}$. We assume that the class-conditional distribution $p_k^{\text{te}}(\boldsymbol{x}|\boldsymbol{y}) = p_k^{\text{tr}}(\boldsymbol{x}|\boldsymbol{y}) := p(\boldsymbol{x}|\boldsymbol{y})$ remains the same for all nodes $k$. This is a common assumption and holds when label shifts primarily affect labels' prior distribution of the labels $p(\boldsymbol{y})$ rather than the underlying feature distribution given the labels, e.g., when features that are generated given a label remains constant (Zadrozny, 2004; Huang et al., 2006; Sugiyama et al., 2007). Note that $p_k^{\text{tr}}(\boldsymbol{y})$ and $p_k^{\text{te}}(\boldsymbol{y})$ can be arbitrarily different, which gives rise to intra- and inter-node *label shifts* (Zadrozny, 2004; Huang et al., 2006; Sugiyama et al., 2007; Garg et al., 2023).

In this multi-node environment, the aim is to find an unbiased estimate of the overall *true risk* minimizer across multiple nodes under both intra-node and inter-node *label shifts*. Specifically, we aim to find a hypothesis $h_{\boldsymbol{w}} \in \mathcal{H} : \mathcal{X} \to \mathcal{Y}$, represented by a neural network parameterized by $\boldsymbol{w}$, such that $h_{\boldsymbol{w}}(\boldsymbol{x})$ provides a good approximation of the label $\boldsymbol{y} \in \mathcal{Y}$ corresponding to a new sample $\boldsymbol{x} \in \mathcal{X}$ drawn from the aggregated *test* data. Let $\ell : \mathcal{X} \times \mathcal{Y} \to \mathbb{R}_+$ denote a loss function. Node $k$ aims to learn a hypothesis $h_{\boldsymbol{w}}$ that minimizes its true (expected) risk:

$$R_k(h_{\boldsymbol{w}}) = \mathbb{E}_{(\boldsymbol{x},\boldsymbol{y}) \sim p_k^{\text{te}}(\boldsymbol{x},\boldsymbol{y})}[\ell(h_{\boldsymbol{w}}(\boldsymbol{x}), \boldsymbol{y})]. \qquad \text{(Local Risk)}$$

---

[2]Sets and scalars are represented by calligraphic and standard fonts, respectively. We use $[m]$ to denote $\{1, \ldots, m\}$ for an integer $m$. We use $\lesssim$ to ignore terms up to constants and logarithmic factors. We use $\mathbb{E}[\cdot]$ to denote the expectation and $\|\cdot\|$ to represent the Euclidean norm of a vector. We use lower-case bold font to denote vectors.

We now modify the classical ERM and formulate IW-ERM to find a predictor that minimizes the overall true risk over all nodes under label shifts:

$$\min_{h_{\boldsymbol{w}} \in \mathcal{H}} \sum_{k=1}^{K} \frac{1}{n_k^{\text{tr}}} \sum_{i=1}^{n_k^{\text{tr}}} \frac{\sum_{j=1}^{K} p_j^{\text{te}}(\boldsymbol{y}_{k,i}^{\text{tr}})}{p_k^{\text{tr}}(\boldsymbol{y}_{k,i}^{\text{tr}})} \ell(h_{\boldsymbol{w}}(\boldsymbol{x}_{k,i}^{\text{tr}}), \boldsymbol{y}_{k,i}^{\text{tr}}), \qquad \text{(IW-ERM)}$$

where $n_k^{\text{tr}}$ is the number of training samples at node $k$.

To incorporate our VRLS importance ratio estimation method into the IW-ERM framework, we replace the ratio term $\frac{\sum_{j=1}^{K} p_j^{\text{te}}(\boldsymbol{y}_{k,i}^{\text{tr}})}{p_k^{\text{tr}}(\boldsymbol{y}_{k,i}^{\text{tr}})}$ with our estimated importance ratios. This modification aims to align the empirical risk minimization with the true risk minimization over all nodes. We formalize the convergence of this approach in Proposition 4.1.

**Proposition 4.1.** *Under the label shift setting described in Section 1, equation IW-ERM is consistent and the learned function $h_{\boldsymbol{w}}$ converges in probability towards the optimal function that minimizes the overall true risk across nodes, $\sum_{k=1}^{K} R_k$.*

*Proof.* Due to space limitations, the proof is provided in Appendix C. Convergence in probability is established by applying the law of large numbers following (Shimodaira, 2000)[Section 3] and (Sugiyama et al., 2007)[Section 2.2]. □

## 5 RATIO ESTIMATION BOUNDS AND CONVERGENCE RATES

In this section, we present bounds on ratio estimation and convergence rates for the finite sample errors incurred during the estimation, as further discussed in Appendices E, F. In practice, we only have access to a finite number of labeled training samples, $\{(\boldsymbol{x}_i, \boldsymbol{y}_i)\}_{i=1}^{n^{\text{tr}}}$, and a finite number of unlabeled test samples, $\{\boldsymbol{x}_j\}_{j=1}^{n^{\text{te}}}$. These samples serve to compute the following estimates:

$$\hat{\boldsymbol{\theta}}_{n^{\text{tr}}} = \arg\min_{\boldsymbol{\theta} \in \Theta} \frac{1}{n^{\text{tr}}} \sum_{i=1}^{n^{\text{tr}}} \left( \ell_{CE}(f_{\boldsymbol{\theta}}(\boldsymbol{x}_i), \boldsymbol{y}_i) + \zeta \Omega(f_{\boldsymbol{\theta}}) \right),$$

$$\text{and } \hat{\boldsymbol{r}}_{n^{\text{te}}} = \arg\max_{\boldsymbol{r} \in \mathbb{R}_+^m} \frac{1}{n^{\text{te}}} \sum_{j=1}^{n^{\text{te}}} \log(f_{\hat{\boldsymbol{\theta}}_{n^{\text{tr}}}}(\boldsymbol{x}_j)^{\top} \boldsymbol{r}).$$

We will show that the errors of these estimates can be controlled. The following assumptions are necessary to establish our results.

**Assumption 5.1** (Boundedness). *The data and the parameter space $\Theta$ are bounded, i.e, there exists $b_{\mathcal{X}}, b_{\Theta} > 0$ such that*

$$\forall \boldsymbol{x} \in \mathcal{X}, \; \|\boldsymbol{x}\|_2 \leq b_{\mathcal{X}} \qquad \text{and} \qquad \forall \boldsymbol{\theta} \in \Theta, \; \|\boldsymbol{\theta}\|_2 \leq b_{\Theta}.$$

**Assumption 5.2** (Calibration). *Let $\boldsymbol{\theta}^{\star}$ be as defined in Equation (4). There exists $\mu > 0$ such that*

$$\mathbb{E}\left[f_{\boldsymbol{\theta}^{\star}}(\boldsymbol{x}) f_{\boldsymbol{\theta}^{\star}}(\boldsymbol{x})^{\top}\right] \succeq \mu \boldsymbol{I}_m.$$

The calibration Assumption 5.2 first appears in (Garg et al., 2020). It is necessary for the ratio estimation procedure to be consistent and we refer the reader to Section 4.3 of Garg et al. (2020) for more details. We further need Assumption 5.1 because, unlike (Garg et al., 2020), the empirical estimator $\hat{\boldsymbol{r}}_{n^{\text{te}}}$ is estimated using another estimator $\hat{\boldsymbol{\theta}}_{n^{\text{tr}}}$. Uniform bounds are therefore needed to control finite sample error as we cannot directly apply concentration inequalities, as is done in the proof of (Garg et al., 2020, Lemma 3), since we do not have independence of the terms appearing in the empirical sums. We nonetheless prove a similar result in the following theorem.

**Theorem 5.1** (Ratio Estimation Error Bound). *Let $\delta \in (0,1)$ and $\mathcal{F} := \{\boldsymbol{x} \mapsto \boldsymbol{r}^{\top} f_{\boldsymbol{\theta}}(\boldsymbol{x}), \; (\boldsymbol{r}, \boldsymbol{\theta}) \in \mathcal{R} \times \Theta\}$. Under Assumptions 5.1-5.2, there exist constants $L > 0, B > 0$ such that with probability at least $1 - \delta$:*

$$\|\hat{\boldsymbol{r}}_{n^{te}} - \boldsymbol{r}_{f^{\star}}\|_2 \leq \frac{2}{\mu p_{\min}} \left( \frac{4}{\sqrt{n^{te}}} Rad(\mathcal{F}) + 4B \sqrt{\frac{\log(4/\delta)}{n^{te}}} \right) + \frac{4L}{\mu p_{\min}} \mathbb{E}\left[\|\boldsymbol{\theta} - \boldsymbol{\theta}^{\star}\|_2\right]. \qquad (5)$$

*Here,* $p_{\min} = \min_y p(y)$ *and*

$$Rad(\mathcal{F}) = \frac{1}{\sqrt{n^{tr}}} \mathbb{E}_{\sigma_1,\ldots,\sigma} \left[ \sup_{(\boldsymbol{r},\boldsymbol{\theta}) \in \mathcal{R} \times \Theta} \left| \sum_{i=1}^{n^{tr}} \sigma_i \boldsymbol{r}^\top f_{\boldsymbol{\theta}}(\boldsymbol{x}_i) \right| \right], \tag{6}$$

*where* $\sigma_1, \ldots, \sigma$ *are Rademacher variables uniformly chosen from* $\{-1, 1\}$.

*Proof.* The proof of Theorem 5.1 is provided in Appendix E. The Rademacher complexity appearing in the bound will depend on the function class chosen for $f$. Moreover as regularization often encourages lower complexity functions, this complexity can be reduced because of the presence of the regularization term in the estimation of $\boldsymbol{\theta}$ in our setting. □

By estimating the ratios locally and incorporating them into local losses, the properties of the modified loss with respect to neural network parameters $\boldsymbol{w}$ remain unchanged, with data-dependent parameters like Lipschitz constants scaled linearly by $r_{\max}$. Our approach trains the predictor using only local data, ensuring IW-ERM with VRLS retains the same privacy guarantees as baseline ERM-solvers. Communication involves only the marginal label distribution, adding negligible overhead, as it is far smaller than model parameters and requires just one round of communication. Overall, importance weighting does not impact communication guarantees during optimization.

**Theorem 5.2** (Convergence-communication). *Let* $\max_{\boldsymbol{y} \in \mathcal{Y}} \sup_f r_f(\boldsymbol{y}) = r_{\max}$. *Suppose Algorithm 2, e.g., IW-ERM with VRLS for multi-node environment, is run for* $T$ *iterations. Then Algorithm 2 achieves a convergence rate of* $\mathcal{O}(r_{\max} h(T))$, *where* $\mathcal{O}(h(T))$ *denotes the rate of ERM-solver baseline without importance weighting. Throughout the course of optimization, Algorithm 2 has the same overall communication guarantees as the baseline.*

In the following, we establish tight convergence rates and communication guarantees for IW-ERM with VRLS in a broad range of importance optimization settings including upper and lower bounds for convex optimization (Theorems 5.3- 5.4), second-order differentiability, composite optimization with proximal operator (Theorem F.3), optimization with adaptive step-sizes, and nonconvex optimization (Theorems F.1- F.2), along the lines of e.g., (Woodworth et al., 2020; Haddadpour et al., 2021; Glasgow et al., 2022; Liu et al., 2023; Hu & Huang, 2023; Wu et al., 2023; Liu et al., 2023).

**Assumption 5.3** (Convex and Smooth). *1) A minimizer* $\boldsymbol{w}^\star$ *exists with bounded* $\|\boldsymbol{w}^\star\|_2$; *2) The* $\ell \circ h_{\boldsymbol{w}}$ *is* $\beta$-smoothness and convex w.r.t. $\boldsymbol{w}$; *3) The stochastic gradient* $\boldsymbol{g}(\boldsymbol{w}) = \widetilde{\nabla}_{\boldsymbol{w}} \ell(h_{\boldsymbol{w}})$ *is unbiased, i.e.,* $\mathbb{E}[\boldsymbol{g}(\boldsymbol{w})] = \nabla_{\boldsymbol{w}} \ell(h_{\boldsymbol{w}})$ *for any* $\boldsymbol{w} \in \mathcal{W}$ *with bounded variance* $\mathbb{E}[\|\boldsymbol{g}(\boldsymbol{w}) - \nabla_{\boldsymbol{w}} \ell(h_{\boldsymbol{w}})\|_2^2]$.

For convex and smooth optimization, we establish convergence rates for IW-ERM with VRLS and local updating along the lines of e.g., (Woodworth et al., 2020, Theorem 2).

**Theorem 5.3** (Upper Bound for Convex and Smooth). *Let* $D = \|\boldsymbol{w}_0 - \boldsymbol{w}^\star\|$, $\tau$ *denote the number of local steps (number of stochastic gradients per round of communication per node),* $R$ *denote the number of communication rounds, and* $\max_{\boldsymbol{y} \in \mathcal{Y}} \sup_f r_f(\boldsymbol{y}) = r_{\max}$. *Under Assumption 5.3, suppose Algorithm 2 with* $\tau$ *local updates is run for* $T = \tau R$ *total stochastic gradients per node with an optimally tuned and constant step-size. Then we have the following upper bound:*

$$\mathbb{E}[\ell(h_{\boldsymbol{w}_T}) - \ell(h_{\boldsymbol{w}^\star})] \lesssim \frac{r_{\max} \beta D^2}{\tau R} + \frac{(r_{\max} \beta D^4)^{1/3}}{(\sqrt{\tau} R)^{2/3}} + \frac{D}{\sqrt{K \tau R}}. \tag{7}$$

**Assumption 5.4** (Convex and Second-order Differentiable). *1) The* $\ell(h_{\boldsymbol{w}}(\boldsymbol{x}), \boldsymbol{y})$ *is* $\beta$-smoothness and convex w.r.t. $\boldsymbol{w}$ *for any* $(\boldsymbol{x}, y)$; *2) The stochastic gradient* $\boldsymbol{g}(\boldsymbol{w}) = \widetilde{\nabla}_{\boldsymbol{w}} \ell(h_{\boldsymbol{w}})$ *is unbiased, i.e.,* $\mathbb{E}[\boldsymbol{g}(\boldsymbol{w})] = \nabla_{\boldsymbol{w}} \ell(h_{\boldsymbol{w}})$ *for any* $\boldsymbol{w} \in \mathcal{W}$ *with bounded variance* $\mathbb{E}[\|\boldsymbol{g}(\boldsymbol{w}) - \nabla_{\boldsymbol{w}} \ell(h_{\boldsymbol{w}})\|_2^2]$.

**Theorem 5.4** (Lower Bound for Convex and Second-order Differentiable). *Let* $D = \|\boldsymbol{w}_0 - \boldsymbol{w}^\star\|$, $\tau$ *denote the number of local steps,* $R$ *denote the number of communication rounds, and* $\max_{\boldsymbol{y} \in \mathcal{Y}} \sup_f r_f(\boldsymbol{y}) = r_{\max}$. *Under Assumption 5.4, suppose Algorithm 2 with* $\tau$ *local updates is run for* $T = \tau R$ *total stochastic gradients per node with a tuned and constant step-size. Then we have the following lower bound:*

$$\mathbb{E}[\ell(h_{\boldsymbol{w}_T}) - \ell(h_{\boldsymbol{w}^\star})] \gtrsim \frac{r_{\max} \beta D^2}{\tau R} + \frac{(r_{\max} \beta D^4)^{1/3}}{(\sqrt{\tau} R)^{2/3}} + \frac{D}{\sqrt{K \tau R}}. \tag{8}$$

We finally establish high-probability convergence bounds for IW-ERM with VRLS along the lines of e.g., (Liu et al., 2023, Theorem 4.1). To show the impact of importance weighting on convergence rate decoupled from the impact of number of nodes and obtain the current SotA *high-probability* bounds for nonconvex optimization, we focus on IW-ERM with $K = 1$.

**Assumption 5.5** (Sub-Gaussian Noise). *1) A minimizer $\boldsymbol{w}^\star$ exists; 2) The stochastic gradients $\boldsymbol{g}(\boldsymbol{w}) = \widetilde{\nabla}_{\boldsymbol{w}} \ell(h_{\boldsymbol{w}})$ is unbiased, i.e., $\mathbb{E}[\boldsymbol{g}(\boldsymbol{w})] = \nabla_{\boldsymbol{w}} \ell(h_{\boldsymbol{w}})$ for any $\boldsymbol{w} \in \mathcal{W}$; 3) The noise $\|\boldsymbol{g}(\boldsymbol{w}) - \nabla_{\boldsymbol{w}} \ell(h_{\boldsymbol{w}})\|_2$ is $\sigma$-sub-Gaussian (Vershynin, 2018).*

**Theorem 5.5** (High-probability Bound for Nonconvex Optimization). *Let $\delta \in (0, 1)$ and $T \in \mathbb{Z}_+$. Let $K = 1$ and $\max_{\boldsymbol{y} \in \mathcal{Y}} \sup_f r_f(\boldsymbol{y}) = r_{\max}$. Under Assumption 5.5 and $\beta$-smoothness of nonconvex $\ell \circ h_{\boldsymbol{w}}$, suppose IW-ERM is run for $T$ iterations with a step-size $\min\left\{\frac{1}{r_{\max}\beta}, \sqrt{\frac{1}{\sigma^2 r_{\max}\beta T}}\right\}$. Then with probability $1 - \delta$, gradient norm squareds satisfy:*

$$\frac{1}{T} \sum_{t=1}^{T} \|\nabla_{\boldsymbol{w}} \ell(h_{\boldsymbol{w}_t})\|_2^2 = O\left(\sigma \sqrt{\frac{r_{\max}\beta}{T}} + \frac{\sigma^2 \log(1/\delta)}{T}\right). \tag{9}$$

*Proof.* We note that importance ratios do not depend on the model parameters $\boldsymbol{w}$ and the Lipschitz and smoothness constants for $\ell \circ h_{\boldsymbol{w}}$ w.r.t. $\boldsymbol{w}$ are scaled by $r_{\max}$. The rest of the proof follows the arguments of (Liu et al., 2023, Theorem 4.1). $\square$

Theorem 5.5 shows that when the stochastic gradients are too noisy $\sigma = \Omega(\sqrt{r_{\max}\beta}/\log(1/\delta))$ such that the second term in the rate dominates, then importance weighting does not have any negative impact on the convergence rate.

## 6 EXPERIMENTS

The experiments are divided into two main parts: evaluating VRLS's performance on a single node focusing on intra-node label shifts, and extending it to multi-node distributed learning scenarios with 5, 100, and 200 nodes. In the multi-node cases, we account for both inter-node and intra-node label shifts. Further experimental details, results, and discussions are provided in Appendix J.

**Importance ratio estimation.** We begin by evaluating VRLS on the MNIST (LeCun et al., 1998) and CIFAR-10 (Krizhevsky) datasets in a single-node setting. Following the common experimental setup in the literature (Lipton et al., 2018), we simulate the test dataset using a Dirichlet distribution with varying $\alpha$ parameters. In this context, a higher $\alpha$ value indicates smoother transitions in the label distribution, while lower values reflect more abrupt shifts. The training dataset is uniformly distributed across all classes. Initially, using a sample size of 5,000, we investigate 20 $\alpha$ values within the range $[10^{-1}, 10^1]$. Next, we fix $\alpha$ at either 1.0 or 0.1 and explore 50 different sample sizes ranging from 200 to 10,000. For each experiment, we run 100 trials and compute the mean squared error (MSE) between the true ratios and the estimated ratios. A two-layer MLP is used for MNIST, while ResNet-18 (He et al., 2016) is applied for CIFAR-10.

| Method | Avg.accuracy |
|---|---|
| **Our IW-ERM** | **0.7520 ± 0.0209** |
| Our IW-ERM (small) | 0.7376 ± 0.0099 |
| FedAvg | 0.5472 ± 0.0297 |
| FedBN | 0.5359 ± 0.0306 |
| FedProx | 0.5606 ± 0.0070 |
| SCAFFOLD | 0.5774 ± 0.0036 |
| Upper Bound | 0.8273 ± 0.0041 |

Table 2: We utilize LeNet on Fashion MNIST to address label shifts across 5 nodes. For the baseline methods—FedAvg, FedBN, FedProx, and SCAFFOLD—we run 15,000 iterations, while both the Upper Bound (IW-ERM with true ratios) and our IW-ERM with VRLS are limited to 5,000 iterations. Notably, we employ a simple MLP with dropout for training the predictor. The model labeled *Our IW-ERM (small)* refers to our approach where the black-box predictor is trained using only 10% of the available training data, balancing computational efficiency with competitive performance.

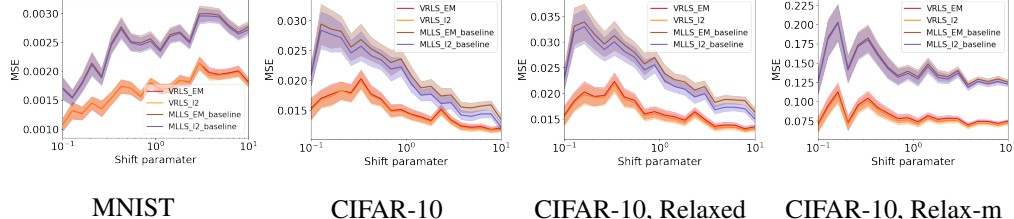

| MNIST | CIFAR-10 | CIFAR-10, Relaxed | CIFAR-10, Relax-m |

Figure 1: MSE analysis across different datasets and settings for VRLS (ours) compared to baselines, focusing on **shift parameter** ($\alpha$) experiments. These subfigures include results from MNIST, CIFAR-10, and relaxed label shift, illustrating the consistent superiority of VRLS. In the **'relaxed'** setting, Gaussian blur (kernel size: 3; $\sigma$: 0.1–0.5) and brightness adjustment (factor: ±0.1) are applied with a 30% probability to introduce real-world variability. In the **'relax-m'** scenario, augmentations are applied with a 50% probability, with Gaussian blur ($\sigma$: 0.1–0.7) and brightness (factor: ±0.2).

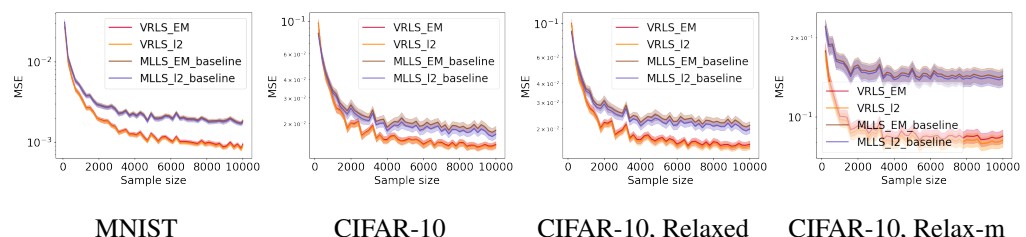

| MNIST | CIFAR-10 | CIFAR-10, Relaxed | CIFAR-10, Relax-m |

Figure 2: MSE analysis across different datasets and settings for VRLS (ours) compared to baselines, focusing on **sample size** experiments. These subfigures include results from MNIST, CIFAR-10, and relaxed label shift conditions, highlighting VRLS's superior performance across varying test set sizes.

Table 3: We deploy ResNet-18 on CIFAR-10 to address label shifts across 5 nodes. The predictor is also a ResNet-18, ensuring consistency with the single-node scenario. For a fair comparison, we limit IW-ERM with VRLS and the true ratios to 5,000 iterations, while FedAvg and FedBN are run for 10,000 iterations. Detailed results are provided in Table 7.

| CIFAR-10 | **Our IW-ERM** | FedAvg | FedBN | Upper Bound |
|---|---|---|---|---|
| Avg. accuracy | **$0.5640 \pm 0.0241$** | $0.4515 \pm 0.0148$ | $0.4263 \pm 0.0975$ | $0.5790 \pm 0.0103$ |

Table 4: We present the average node accuracies from the CIFAR-10 target shift experiment conducted with 100 and 200 nodes, where 5 nodes are randomly sampled to participate in each training round. Our IW-ERM with VRLS is run for 5,000 and 10,000 iterations, respectively, while both FedAvg and FedBN are run for 10,000 iterations each.

| CIFAR-10 | **Our IW-ERM** | FedAvg | FedBN |
|---|---|---|---|
| Avg. accuracy (100 nodes) | **0.5354** | 0.3915 | 0.1537 |
| Avg. accuracy (200 nodes) | **0.6216** | 0.5942 | 0.1753 |

Figure 1 and Figure 2 compares our proposed VRLS with baselines (Garg et al., 2020; Saerens et al., 2002) under label shifts. MLLS_L2 refers to the MLLS method using convex optimization via SGD (Garg et al., 2020), while MLLS_EM employs the same objective function but is optimized using the EM algorithm (Saerens et al., 2002). Our proposed VRLS is optimized in a similar manner, resulting in VRLS_L2 and VRLS_EM, as shown in the figure. Our method consistently achieves lower MSE across different label shift intensities ($\alpha$) and test sample sizes on both datasets. Notably, our importance ratio estimation experiments align with the error bound in Theorem 5.1, demonstrating that increasing the number of test samples improves estimation error at a rate proportional to the square root of the sample size. Additionally, the regularization term constrains the parameter space and reduces Rademacher complexity, leading to smoother predictions and improved model calibration, as supported by Section S2 in (Guo et al., 2017a). Both of them contribute to reduced estimation error.

We also tested importance ratio estimation under relaxed label shift conditions and found VRLS to exhibit greater robustness (see Appendix J.2 for detailed settings). Although this assumption holds broader potential for real-world applications, its precise alignment with real-world datasets requires further investigation—an important direction for future research that extends beyond the scope of this work.

**Distributed learning settings.** We apply VRLS in a distributed learning context, addressing both intra- and inter-node label shifts. The initial experiments involve 5 nodes, using predefined label distributions on Fashion MNIST (Xiao et al., 2017) and CIFAR-10, as shown in Tables 8- 9 in Appendix J.

We employ a simple MLP with dropout as the predictor for Fashion MNIST. For global training with IW-ERM, LeNet (LeCun et al., 1998) is used on Fashion MNIST, and ResNet-18 (Ramezani-Kebrya et al., 2023b) on CIFAR-10. All experiments are run with three random seeds, reporting the average accuracy across nodes. We compare IW-ERM with VRLS against baseline methods, including FedAvg (McMahan et al., 2017), FedBN (Li et al., 2021b), FedProx (Li et al., 2020), and SCAFFOLD (Karimireddy et al., 2020a), as well as IW-ERM with true ratios serving as an upper bound. Hyperparameters are kept consistent with those in (McMahan et al., 2017; Li et al., 2021b; Ramezani-Kebrya et al., 2023b).

Each node's stochastic gradients are computed with a batch size of 64 and aggregated using the Adam optimizer. All experiments are run on a single GPU within an internal cluster. Both MLLS and VRLS use identical hyperparameters and training epochs for CIFAR-10 and Fashion MNIST, stopping once the classification loss reaches a predefined threshold on MNIST. We also conduct experiments with 100 and 200 nodes on CIFAR-10, where five nodes are randomly sampled each iteration to simulate more realistic distributed learning. In this case, IW-ERM with true ratios does not act as the upper bound due to the stochastic node sampling. The experiment is run once, and average accuracy across nodes is reported, with label distribution shown in Table 10 in Appendix J. Despite FedBN's reported slow convergence (Ramezani-Kebrya et al., 2023b), we maintain 15,000 and 10,000 iterations for FedAvg and FedBN on Fashion MNIST and CIFAR-10, respectively, for fair comparison. However, IW-ERM is limited to 5,000 iterations using both true and estimated ratios due to faster convergence.

As shown in Table 2, IW-ERM achieves over 20% higher average accuracy than all baselines on Fashion MNIST, with only a third of the iterations. Notably, even with just 10% of the training data in the first round of global training, the performance remains comparable, demonstrating reduced training complexity. This improvement is attributed to the theoretical benefits of IW-ERM, the robustness of density estimation, and the fact that the aggregation of importance ratios reduces reliance on any single local estimate. Similarly, Table 3 shows that IW-ERM approaches the upper bound on CIFAR-10, outperforming the baselines. Individual node accuracies are detailed in Tables 6-7 in Appendix J. In the 100-node scenario, IW-ERM continues to demonstrate superior performance, requiring only half the iterations, as shown in Table 4. It is important to note that using true ratios does not equate to IW-ERM, given the stochasticity of node selection during training.

## 7 CONCLUSIONS AND LIMITATIONS

We propose VRLS to address label shift in distributed learning. Paired with IW-ERM, VRLS improves intra- and inter-node label shifts in multi-node settings. Empirically, VRLS consistently outperforms MLLS-based baselines, and IW-ERM with VRLS exceeds all multi-node learning baselines. Theoretical bounds further strengthen our method's foundation. Future work will explore estimating ratios by relaxing the strict class-conditional assumption and optimizing IW-ERM to reduce time complexity while ensuring scalability and practicality in real-world distributed learning.

## ETHICS STATEMENT

No ethical approval was needed as no human subjects were involved. All authors fully support the content and findings.

## REPRODUCIBILITY STATEMENT

We ensured reproducibility with publicly available datasets (MNIST, CIFAR-10) and standard models (e.g., ResNet-18). Links to datasets, code, and configurations will be provided upon camera-ready submission. Experiments were run on NVIDIA 3090, A100 GPUs, and Google Colab, with average results and variances reported across multiple trials.

## ACKNOWLEDGMENTS

The authors would like to thank Leello Tadesse Dadi and Thomas Pethick for helpful discussions. This work was supported by the Research Council of Norway (RCN) through its Centres of Excellence scheme, Integreat: Norwegian Centre for knowledge-driven machine learning, project number 332645. The work of Changkyu Choi was funded by RCN under grant 309439. The work of Volkan Cevher was supported by Hasler Foundation Program: Hasler Responsible AI (project number 21043), the Army Research Office which was accomplished under Grant Number W911NF-24-1-0048, and the Swiss National Science Foundation (SNSF) under grant number 200021_205011. The computations were performed in part on resources provided by Sigma2 - the National Infrastructure for High-Performance Computing and Data Storage in Norway.

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

The Appendix part is organized as follows:

- All related work are provided in Appendix A.
- Additional details of prior work of BBSE and MLLS are in  Appendix B.
- Mathematical proof for label shifts with multiple nodes and IW-ERM is given in Appendix C.
- General algorithmic description is in Appendix D.
- Proof of Theorem 5.1 is in Appendix E.
- Proof of Theorem 5.2 and Convergence-Communication-Privacy guarantees for IW-ERM in Equation (IW-ERM) are provided in Appendix F.
- Complexity analysis is in Appendix G.
- Mathematical notations are summarized in Appendix H.
- Limitations are discussed in Appendix I.
- Additional experiments and experimental details are provided in Appendix J.

## A  RELATED WORK

In the context of distributed learning with label shifts, importance ratio estimation is tackled either by solving a linear system as in (Lipton et al., 2018; Azizzadenesheli et al., 2019) or by minimizing distribution divergence as in (Garg et al., 2020). In this section, we overview complete related work.

**Federated learning (FL).**    Much of the current research in FL predominantly centers around the minimization of empirical risk, operating under the assumption that each node maintains the same training/test data distribution (Kairouz et al., 2021). Prominent methods in FL include FedAvg (McMahan et al., 2017), FedBN (Li et al., 2021b), FedProx (Li et al., 2020) and SCAFFOLD (Karimireddy et al., 2020a). FedAvg and its variants such as (Huang et al., 2021; Karimireddy et al., 2020b) have been the subject of thorough investigation in optimization literature, exploring facets such as communication efficiency, node participation, and privacy assurance (Faghri et al., 2020; Ramezani-Kebrya et al., 2021; 2023a;b). Subsequent work, such as the study by de Luca et al. (2022), explores Federated Domain Generalization and introduces data augmentation to the training. This model aims to generalize to both in-domain datasets from participating nodes and an out-of-domain dataset from a non-participating node. Additionally, Gupta et al. (2022) introduces FL Games, a game-theoretic framework designed to learn causal features that remain invariant across nodes. This is achieved by employing ensembles over nodes' historical actions and enhancing local computation, under the assumption of consistent training/test data distribution across nodes. The existing strategies to address statistical heterogeneity across nodes during training primarily rely on heuristic-based personalization methods, which currently lack theoretical backing in statistical learning (Smith et al., 2017; Khodak et al., 2019; Li et al., 2021a). In contrast, we aim to minimize overall test error amid both intra-node and inter-node distribution shifts, a situation frequently observed in real-world scenarios. Techniques ensuring communication efficiency, robustness, and secure aggregations serve as complementary.

**Importance ratio estimation**    Classical Empirical Risk Minimization (ERM) seeks to minimize the expected loss over the training distribution using finite samples. When faced with distribution shifts, the goal shifts to minimizing the expected loss over the target distribution, leading to the development of Importance-Weighted Empirical Risk Minimization (IW-ERM)(Shimodaira, 2000; Sugiyama et al., 2006; Byrd & C. Lipton, 2019; Fang et al., 2020). Shimodaira (2000) established that the IW-ERM estimator is asymptotically unbiased. Moreover, Ramezani-Kebrya et al. (2023b) introduced FTW-ERM, which integrates importance ratio estimation.

**Label shift and MLLS family**    For theoretical analysis, the conditional distribution $p(\boldsymbol{x}|\boldsymbol{y})$ is held strictly constant across all distributions (Lipton et al., 2018; Garg et al., 2020; Saerens et al., 2002). Both BBSE (Lipton et al., 2018) and RLLS (Azizzadenesheli et al., 2019) designate a discrete latent space $\boldsymbol{z}$ and introduce a confusion matrix-based estimation method to compute the ratio $\boldsymbol{w}$ by solving a linear system (Saerens et al., 2002; Lipton et al., 2018). This approach is straightforward and has been proven consistent, even when the predictor is not calibrated. However, its subpar performance is attributed to the information loss inherent in the confusion matrix (Garg et al., 2020).

Consequently, MLLS (Garg et al., 2020) introduces a continuous latent space, resulting in a significant enhancement in estimation performance, especially when combined with a post-hoc calibration method (Shrikumar et al., 2019). It also provides a consistency guarantee with a canonically calibrated predictor. This EM-based MLLS method is both concave and can be solved efficiently.

**Discrepancy Measure**    In information theory and statistics, discrepancy measures play a critical role in quantifying the differences between probability distributions. One such measure is the Bregman Divergence (Banerjee et al., 2005), defined as

$$D_\phi(\boldsymbol{x}\|\boldsymbol{y}) = \phi(\boldsymbol{x}) - \phi(\boldsymbol{y}) - \langle \nabla\phi(\boldsymbol{y}), \boldsymbol{x} - \boldsymbol{y} \rangle,$$

which encapsulates the difference between the value of a convex function $\phi$ at two points and the value of the linear approximation of $\phi$ at one point, leveraging the gradient at another point.

Discrepancy measures are generally categorized into two main families: Integral Probability Metrics (IPMs) and $f$-divergences. IPMs, including Maximum Mean Discrepancy (Gretton et al., 2012) and Wasserstein distance (Villani, 2009), focus on distribution differences $P - Q$. In contrast, $f$-divergences, such as KL-divergence (Kullback & Leibler, 1951) and Total Variation distance, operate

on ratios $P/Q$ and do not satisfy the triangular inequality. Interconnections and variations between these families are explored in studies like $(f, \Gamma)$-Divergences (Birrell et al., 2022), which interpolate between $f$-divergences and IPMs, and research outlining optimal bounds between them (Agrawal & Horel, 2020).

MLLS (Garg et al., 2020) employs $f$-divergence, notably the KL divergence, which is not a metric as it doesn't satisfy the triangular inequality, and requires distribution $P$ to be absolutely continuous with respect to $Q$. Concerning IPMs, while MMD is reliant on a kernel function, it can suffer from the curse of dimensionality when faced with high-dimensional data. On the other hand, the Wasserstein distance can be reformulated using Kantorovich-Rubinstein duality (Dedecker et al., 2006; Arjovsky et al., 2017) as a maximization problem subject to a Lipschitz constrained function $f : \mathbb{R}^d \to \mathbb{R}$.

## B BBSE AND MLLS FAMILY

In this section, we summarize the contributions of BBSE (Lipton et al., 2018) and MLLS (Garg et al., 2020). Our objective is to estimate the ratio $p^{\text{te}}(y)/p^{\text{tr}}(y)$. We consider a scenario with $m$ possible label classes, where $y = c$ for $c \in [m]$. Let $\boldsymbol{r}^{\star} = [r_1^{\star}, \ldots, r_m^{\star}]^{\top}$ represent the true ratios, with each $r_c^{\star}$ defined as $r_c^{\star} = \frac{p^{\text{te}}(y=c)}{p^{\text{tr}}(y=c)}$ (Garg et al., 2020). We then define a family of distributions over $\mathcal{Z}$, parameterized by $\boldsymbol{r} = [r_1, \ldots, r_m]^{\top} \in \mathbb{R}^m$, where $r_c$ is the $c$-th element of the ratio vector.

$$p_{\boldsymbol{r}}(\boldsymbol{z}) := \sum_{c=1}^{m} p^{\text{te}}(\boldsymbol{z}|y=c) \cdot p^{\text{tr}}(y=c) \cdot r_c \tag{10}$$

Here, $r_c \geq 0$ for $c \in [m]$ and $\sum_{c=1}^{m} r_c \cdot p^{\text{tr}}(y=c) = \sum_{c=1}^{m} p^{\text{te}}(y=c) = 1$ as constraints. When $\boldsymbol{r} = \boldsymbol{r}^{\star}$, e.g., $r_c = r_c^{\star}$ for $c \in [m]$, we have $p_{\boldsymbol{r}}(\boldsymbol{z}) = p_{\boldsymbol{r}^{\star}}(\boldsymbol{z}) = p^{\text{te}}(\boldsymbol{z})$ (Garg et al., 2020). So our task is to find $\boldsymbol{r}$ such that

$$\begin{aligned}
\sum_{c=1}^{m} & p^{\text{te}}(\boldsymbol{z}|y=c) \cdot p^{\text{tr}}(y=c) \cdot r_c \boldsymbol{x} \\
&= \sum_{c=1}^{m} p^{\text{tr}}(\boldsymbol{z}, y=c) \cdot r_c = p^{\text{te}}(\boldsymbol{z})
\end{aligned} \tag{11}$$

Lipton et al. (2018) introduced Black Box Shift Estimation (BBSE) to address this issue. With a pre-trained classifier $f$ for the classification task, BBSE assumes that the latent space $\mathcal{Z}$ is discrete and defines $p(\boldsymbol{z}|\boldsymbol{x}) = \delta_{\arg\max f(\boldsymbol{x})}$, where the output of $f(\boldsymbol{x})$ is a probability vector (or a simplex) over $m$ classes. BBSE estimates $p^{\text{te}}(\boldsymbol{z}|y)$ as a confusion matrix, using both the training and validation data. It calculates $p^{\text{tr}}(y=c)$ from the training set and $p^{\text{te}}(\boldsymbol{z})$ from the test data. The problem then reduces to solving the following equation:

$$\boldsymbol{Aw} = \boldsymbol{B} \tag{12}$$

where $|\mathcal{Z}| = m$, $\boldsymbol{A} \in \mathbb{R}^{m \times m}$ with $A_{jc} = p^{\text{te}}(z=j|y=c) \cdot p^{\text{tr}}(y=c)$, and $\boldsymbol{B} \in \mathbb{R}^m$ with $B_j = p^{\text{te}}(z=j)$ for $c, j \in [m]$.

The estimation of the confusion matrix in terms of $p^{\text{te}}(\boldsymbol{z}|y)$ leads to the loss of calibration information (Garg et al., 2020). Furthermore, when defining $\mathcal{Z}$ as a continuous latent space, the confusion matrix becomes intractable since $\boldsymbol{z}$ has infinitely many values. Therefore, MLLS directly minimizes the divergence between $p^{\text{te}}(\boldsymbol{z})$ and $p_{\boldsymbol{r}}(\boldsymbol{z})$, instead of solving the linear system in Equation (12).

Within the $f$-divergence family, MLLS seeks to find a weight vector $\boldsymbol{r}$ by minimizing the KL-divergence $D_{\text{KL}}(p^{\text{te}}(\boldsymbol{z}), p_{\boldsymbol{r}}(\boldsymbol{z})) = \mathbb{E}_{\text{te}}[\log p^{\text{te}}(\boldsymbol{z})/p_{\boldsymbol{r}}(\boldsymbol{z})]$, for $p_{\boldsymbol{r}}(\boldsymbol{z})$ defined in Equation (10). Leveraging on the properties of the logarithm, this is equivalent to maximizing the log-likelihood: $\boldsymbol{r} := \arg\max_{\boldsymbol{r} \in \mathbb{R}} \mathbb{E}_{\text{te}}[\log p_{\boldsymbol{r}}(\boldsymbol{z})]$. Expanding $p_{\boldsymbol{r}}(\boldsymbol{z})$, we have

$$\begin{aligned}
\mathbb{E}_{\text{te}}[\log p_{\boldsymbol{r}}(\boldsymbol{z})] &= \mathbb{E}_{\text{te}}\left[\log(\sum_{c=1}^{m} p^{\text{tr}}(\boldsymbol{z}, y=c)r_c)\right] \\
&= \mathbb{E}_{\text{te}}\left[\log(\sum_{c=1}^{m} p^{\text{tr}}(y=c \mid \boldsymbol{z})r_c) + \log p^{\text{tr}}(\boldsymbol{z})\right].
\end{aligned} \tag{13}$$

Therefore the unified form of MLLS can be formulated as:

$$\boldsymbol{r} := \arg\max_{\boldsymbol{r} \in \mathbb{R}} \mathbb{E}_{\text{te}}\left[\log(\sum_{c=1}^{m} p^{\text{tr}}(y=c \mid \boldsymbol{z})r_c)\right]. \tag{14}$$

This is a convex optimization problem and can be solved efficiently using methods such as EM, an analytic approach, and also iterative optimization methods like gradient descent with labeled training data and unlabeled test data. MLLS defines the $p(\boldsymbol{z}|\boldsymbol{x})$ as $\delta_{\boldsymbol{x}}$, plugs in the pre-defined $f$ to approximate $p^{\text{tr}}(y|\boldsymbol{x})$ and optimizes the following objective:

$$\boldsymbol{r}_f := \arg\max_{\boldsymbol{r}\in\mathbb{R}} \ell(\boldsymbol{r}, f) := \arg\max_{\boldsymbol{r}\in\mathbb{R}} \mathbb{E}_{\text{te}}\left[\log(f(\boldsymbol{x})^T \boldsymbol{r})\right]. \tag{15}$$

With the Bias-Corrected Calibration (BCT) (Shrikumar et al., 2019) strategy, they adjust the logits $\hat{f}(\boldsymbol{x})$ of $f(\boldsymbol{x})$ element-wise for each class, and the objective becomes:

$$\boldsymbol{r}_f := \arg\max_{\boldsymbol{r}\in\mathbb{R}} \ell(\boldsymbol{r}, f) := \arg\max_{\boldsymbol{r}\in\mathbb{R}} \mathbb{E}_{\text{te}}\left[\log(g \circ \hat{f}(\boldsymbol{x}))^T \boldsymbol{r})\right], \tag{16}$$

where $g$ is a calibration function.

## C  PROOF OF PROPOSITION 4.1

In the following, we consider four typical scenarios under various distribution shifts described in Table 1 and formulate their IW-ERM with a focus on minimizing $R_1$.

### C.1  NO INTRA-NODE LABEL SHIFT

For simplicity, we assume that there are only 2 nodes, but our results can be extended to multiple nodes. This scenario assumes $p_k^{tr}(\boldsymbol{y}) = p_k^{te}(\boldsymbol{y})$ for $k = 1, 2$, but $p_1^{tr}(\boldsymbol{y}) \neq p_2^{tr}(\boldsymbol{y})$. Node 1 aims to learn $h_{\boldsymbol{w}}$ assuming $\frac{p_1^{tr}(\boldsymbol{y})}{p_2^{tr}(\boldsymbol{y})}$ is given. We consider the following IW-ERM that is consistent in minimizing $R_1$:

$$
\min_{h_{\boldsymbol{w}} \in \mathcal{H}} \frac{1}{n_1^{tr}} \sum_{i=1}^{n_1^{tr}} \ell(h_{\boldsymbol{w}}(\boldsymbol{x}_{1,i}^{tr}), \boldsymbol{y}_{1,i}^{tr})
$$
$$
+ \frac{1}{n_2^{tr}} \sum_{i=1}^{n_2^{tr}} \frac{p_1^{tr}(\boldsymbol{y}_{2,i}^{tr})}{p_2^{tr}(\boldsymbol{y}_{2,i}^{tr})} \ell(h_{\boldsymbol{w}}(\boldsymbol{x}_{2,i}^{tr}), \boldsymbol{y}_{2,i}^{tr}). \tag{17}
$$

Here $\mathcal{H}$ is the hypothesis class of $h_{\boldsymbol{w}}$. This scenario is referred to as `No-LS`.

### C.2  LABEL SHIFT ONLY FOR NODE 1

Here we consider label shift only for node 1, i.e., $p_1^{tr}(\boldsymbol{y}) \neq p_1^{te}(\boldsymbol{y})$ and $p_2^{tr}(\boldsymbol{y}) = p_2^{te}(\boldsymbol{y})$. We consider the following IW-ERM:

$$
\min_{h_{\boldsymbol{w}} \in \mathcal{H}} \frac{1}{n_1^{tr}} \sum_{i=1}^{n_1^{tr}} \frac{p_1^{te}(\boldsymbol{y}_{1,i}^{tr})}{p_1^{tr}(\boldsymbol{y}_{1,i}^{tr})} \ell(h_{\boldsymbol{w}}(\boldsymbol{x}_{1,i}^{tr}), \boldsymbol{y}_{1,i}^{tr})
$$
$$
+ \frac{1}{n_2^{tr}} \sum_{i=1}^{n_2^{tr}} \frac{p_1^{te}(\boldsymbol{y}_{2,i}^{tr})}{p_2^{tr}(\boldsymbol{y}_{2,i}^{tr})} \ell(h_{\boldsymbol{w}}(\boldsymbol{x}_{2,i}^{tr}), \boldsymbol{y}_{2,i}^{tr}). \tag{18}
$$

This scenario is referred to as `LS on single`.

### C.3  LABEL SHIFT FOR BOTH NODES

Here we assume $p_1^{tr}(\boldsymbol{y}) \neq p_1^{te}(\boldsymbol{y})$ and $p_2^{tr}(\boldsymbol{y}) \neq p_2^{te}(\boldsymbol{y})$, i.e., label shift for both nodes. The corresponding IW-ERM is the same as Eq. Equation (18). This scenario is referred to as `LS on both`.

Without loss of generality and for simplicity, we set $l = 1$. We consider four typical scenarios under various distribution shifts and formulate their IW-ERM with a focus on minimizing $R_1$. The details of these scenarios are summarized in Table 1.

### C.4  MULTIPLE NODES

Here we consider a general scenario with $K$ nodes. We assume both intra-node and inter-node label shifts by the following IW-ERM:

$$
\min_{h_{\boldsymbol{w}} \in \mathcal{H}} \sum_{k=1}^{K} \frac{1}{n_k^{tr}} \sum_{i=1}^{n_k^{tr}} \frac{p_1^{te}(\boldsymbol{y}_{k,i}^{tr})}{p_k^{tr}(\boldsymbol{y}_{k,i}^{tr})} \ell(h_{\boldsymbol{w}}(\boldsymbol{x}_{k,i}^{tr}), \boldsymbol{y}_{k,i}^{tr}), \tag{19}
$$

This scenario is referred to as `LS on multi`.

For the scenario without intra-node label shift, the IW-ERM in Equation (17) can be expressed as

$$\frac{1}{n_2^{\text{tr}}} \sum_{i=1}^{n_2^{\text{tr}}} \frac{p_1^{\text{tr}}(\boldsymbol{y}_{2,i}^{\text{tr}})}{p_2^{\text{tr}}(\boldsymbol{y}_{2,i}^{\text{tr}})} \ell(h_{\boldsymbol{w}}(\boldsymbol{x}_{2,i}^{\text{tr}}), \boldsymbol{y}_{2,i}^{\text{tr}})$$

$$\xrightarrow{n_2^{\text{tr}} \to \infty} \mathbb{E}_{p_2^{\text{tr}}(\boldsymbol{x},\boldsymbol{y})} \left[ \frac{p_1^{\text{tr}}(\boldsymbol{y})}{p_2^{\text{tr}}(\boldsymbol{y})} \ell(h_{\boldsymbol{w}}(\boldsymbol{x}), \boldsymbol{y}) \right]$$

$$= \int_{\mathcal{Y}} \frac{p_1^{\text{tr}}(\boldsymbol{y})}{p_2^{\text{tr}}(y)} \mathbb{E}_{p(\boldsymbol{x}|\boldsymbol{y})}[\ell(h_{\boldsymbol{w}}(\boldsymbol{x}), \boldsymbol{y})] p_2^{\text{tr}}(\boldsymbol{y}) d\boldsymbol{y} \tag{20}$$

$$= \int_{\mathcal{Y}} p_1^{\text{tr}}(\boldsymbol{y}) \mathbb{E}_{p(\boldsymbol{x}|\boldsymbol{y})}[\ell(h_{\boldsymbol{w}}(\boldsymbol{x}), \boldsymbol{y})] d\boldsymbol{y}$$

$$= \int_{\mathcal{Y}} p_1^{\text{te}}(\boldsymbol{y}) \mathbb{E}_{p(\boldsymbol{x}|\boldsymbol{y})}[\ell(h_{\boldsymbol{w}}(\boldsymbol{x}), \boldsymbol{y})] d\boldsymbol{y}$$

$$= \mathbb{E}_{p_1^{\text{te}}(\boldsymbol{x},\boldsymbol{y})} [\ell(h_{\boldsymbol{w}}(\boldsymbol{x}), \boldsymbol{y})]$$

$$= R_1(h_{\boldsymbol{w}}).$$

where the second equality holds due to the assumption of the label shift setting and Bayes' theorem: $p(\boldsymbol{x}, \boldsymbol{y}) = p(\boldsymbol{x}|\boldsymbol{y}) \cdot p(\boldsymbol{y})$, and the fourth equality holds by the assumption that $p_1^{\text{tr}}(\boldsymbol{y}) = p_1^{\text{te}}(\boldsymbol{y})$ in the No-LS setting.

For the scenario with label shift only for Node 1 or for both nodes, the IW-ERM in Equation (18) admits

$$\frac{1}{n_2^{\text{tr}}} \sum_{i=1}^{n_2^{\text{tr}}} \frac{p_1^{\text{te}}(\boldsymbol{y}_{2,i}^{\text{tr}})}{p_2^{\text{tr}}(\boldsymbol{y}_{2,i}^{\text{tr}})} \ell(h_{\boldsymbol{w}}(\boldsymbol{x}_{2,i}^{\text{tr}}), \boldsymbol{y}_{2,i}^{\text{tr}}) \tag{21}$$

$$\xrightarrow{n_2^{\text{tr}} \to \infty} \mathbb{E}_{p_2^{\text{tr}}(\boldsymbol{x},\boldsymbol{y})} \left[ \frac{p_1^{\text{te}}(\boldsymbol{y})}{p_2^{\text{tr}}(\boldsymbol{y})} \ell(h_{\boldsymbol{w}}(\boldsymbol{x}), \boldsymbol{y}) \right] \tag{22}$$

$$= \int_{\mathcal{Y}} \frac{p_1^{\text{te}}(y)}{p_2^{\text{tr}}(y)} \mathbb{E}_{p(\boldsymbol{x}|\boldsymbol{y})}[\ell(h_{\boldsymbol{w}}(\boldsymbol{x}), \boldsymbol{y})] p_2^{\text{tr}}(\boldsymbol{y}) d\boldsymbol{y} \tag{23}$$

$$= \int_{\mathcal{Y}} p_1^{\text{te}}(y = \boldsymbol{y}) \mathbb{E}_{p(\boldsymbol{x}|\boldsymbol{y})}[\ell(h_{\boldsymbol{w}}(\boldsymbol{x}), \boldsymbol{y})] d\boldsymbol{y} \tag{24}$$

$$= \mathbb{E}_{p_1^{\text{te}}(\boldsymbol{x},\boldsymbol{y})} [\ell(h_{\boldsymbol{w}}(\boldsymbol{x}), \boldsymbol{y})] \tag{25}$$

$$= R_1(h_{\boldsymbol{w}}). \tag{26}$$

For multiple nodes, let $k \in [K]$. Similarly, we have

$$\frac{1}{n_k^{\text{tr}}} \sum_{i=1}^{n_k^{\text{tr}}} \frac{p_1^{\text{te}}(\boldsymbol{y}_{k,i}^{\text{tr}})}{p_k^{\text{tr}}(\boldsymbol{y}_{k,i}^{\text{tr}})} \ell(h_{\boldsymbol{w}}(\boldsymbol{x}_{k,i}^{\text{tr}}), \boldsymbol{y}_{k,i}^{\text{tr}}) \xrightarrow{n_k^{\text{tr}} \to \infty} R_1(h_{\boldsymbol{w}}). \tag{27}$$

Then we have

$$\sum_{k=1}^{K} \frac{1}{n_k^{\text{tr}}} \sum_{i=1}^{n_k^{\text{tr}}} \frac{p_1^{\text{te}}(\boldsymbol{y}_{k,i}^{\text{tr}})}{p_k^{\text{tr}}(\boldsymbol{y}_{k,i}^{\text{tr}})} \ell(h_{\boldsymbol{w}}(\boldsymbol{x}_{k,i}^{\text{tr}}), \boldsymbol{y}_{k,i}^{\text{tr}}) \xrightarrow{n_1^{\text{tr}}, \ldots, n_K^{\text{tr}} \to \infty} R_1(h_{\boldsymbol{w}}). \tag{28}$$

Note that to solve Equation (19), node 1 needs to estimate $\frac{p_1^{\text{te}}(\boldsymbol{y})}{p_k^{\text{tr}}(\boldsymbol{y})}$ for all nodes $k$ in Equation (19).

The consistency of Equation (IW-ERM), i.e., convergence in probability, is followed the standard arguments in e.g., (Shimodaira, 2000)[Section 3] and (Sugiyama et al., 2007)[Section 2.2] using the law of large numbers.

# D  ALGORITHMIC DESCRIPTION

```python
# Split the training dataset on each node
trainsets = target_shift.split_dataset(trainset.data, trainset.targets,
    node_label_dist_train, transform=transform_train)

# Split the test dataset on each node
testsets = target_shift.split_dataset(testset.data, testset.targets,
    node_label_dist_test, transform=transform_test)

# Initialize K local models (nets) for each node
nets = [initialize_model() for _ in range(node_num)]

# Initialize the estimator for each local model
estimators = [LS_RatioModel(nets[k]) for k in range(node_num)]

# Initialize tensors to store the estimated ratios, values, and marginal
    values for each pair of nodes.
estimated_ratios = torch.zeros(node_num, node_num, nclass)
estimated_values = torch.zeros(node_num, node_num, nclass)
marginal_values = torch.zeros(node_num, nclass)

# Phase 1: Compute the estimated ratios for each node pair (k, j)
for k in range(node_num):
    for j in range(node_num):
        # Perform test on node k using node j's testset
        estimated_ratios[k, j] = estimators[k](testsets[j].data.cpu().
    numpy())

# Phase 2: Compute the marginal values on each node's training set
for i, trainset in enumerate(trainsets):
    marginal_values[i] = marginal(trainset.targets)

# Phase 3: Compute the final estimated values for each node
for k in range(node_num):
    for j in range(node_num):
        estimated_values[k, j] = marginal_values[j] * estimated_ratios[k,
     j]

# Aggregate the estimated values across nodes
aggregated_values = torch.sum(estimated_values, dim=1)

# Compute the final ratios for each node
ratios = (aggregated_values / marginal_values).to(args.device)
```

Listing 1: Our VRLS in distributed learning. It is the implementation of Algorithm 2

# E PROOF OF THEOREM 5.1

*Proof.* Let $H(\boldsymbol{r}, \boldsymbol{\theta}, \boldsymbol{x}) = -\log(f(\boldsymbol{x}, \boldsymbol{\theta})^\top \boldsymbol{r})$. From the strong convexity in Lemma E.7, we have that

$$\|\hat{\boldsymbol{r}}_{n^{\text{te}}} - \boldsymbol{r}_{f^\star}\|_2^2 \leq \frac{2}{\mu p_{\min}} \left( \mathcal{L}_{\boldsymbol{\theta}^\star}(\hat{\boldsymbol{r}}_{n^{\text{te}}}) - \mathcal{L}_{\boldsymbol{\theta}^\star}(\boldsymbol{r}_{f^\star}) \right) \tag{29}$$

Now focusing on the term on the right-hand side, we find by invoking Lemma E.4 that

$$\begin{aligned}
\mathcal{L}_{\boldsymbol{\theta}^\star}&(\hat{\boldsymbol{r}}_{n^{\text{te}}}) - \mathcal{L}_{\boldsymbol{\theta}^\star}(\boldsymbol{r}_{f^\star}) \\
&\leq \mathbb{E}\left[ H(\hat{\boldsymbol{r}}_{n^{\text{te}}}, \hat{\boldsymbol{\theta}}_{n^{\text{tr}}}, \boldsymbol{x}) \right] - \mathbb{E}\left[ H(\boldsymbol{r}_{f^\star}, \hat{\boldsymbol{\theta}}_{n^{\text{tr}}}, \boldsymbol{x}) \right] + 2L\mathbb{E}\left[ \|\hat{\boldsymbol{\theta}}_{n^{\text{tr}}} - \boldsymbol{\theta}^\star\|_2 \right] \\
&= \mathbb{E}\left[ H(\hat{\boldsymbol{r}}_{n^{\text{te}}}, \hat{\boldsymbol{\theta}}_{n^{\text{tr}}}, x) \right] - \frac{1}{n^{\text{te}}} \sum_{j=1}^{n^{\text{te}}} H(\hat{\boldsymbol{r}}_{n^{\text{te}}}, \hat{\boldsymbol{\theta}}_{n^{\text{tr}}}, \boldsymbol{x}_j) + \frac{1}{n^{\text{te}}} \sum_{j=1}^{n^{\text{te}}} H(\hat{\boldsymbol{r}}_n, \hat{\boldsymbol{\theta}}_{n^{\text{tr}}}, \boldsymbol{x}_j) \\
&\qquad - \mathbb{E}\left[ H(\boldsymbol{r}_{f^\star}, \hat{\boldsymbol{\theta}}_{n^{\text{tr}}}, \boldsymbol{x}) \right] + 2L\mathbb{E}\left[ \|\hat{\boldsymbol{\theta}}_{n^{\text{tr}}} - \boldsymbol{\theta}^\star\|_2 \right] \\
&\leq \mathbb{E}\left[ H(\hat{\boldsymbol{r}}_{n^{\text{te}}}, \hat{\boldsymbol{\theta}}_{n^{\text{tr}}}, \boldsymbol{x}) \right] - \frac{1}{n^{\text{te}}} \sum_{j=1}^{n^{\text{te}}} H(\hat{\boldsymbol{r}}_{n^{\text{te}}}, \hat{\boldsymbol{\theta}}_{n^{\text{tr}}}, \boldsymbol{x}_j) + \frac{1}{n^{\text{te}}} \sum_{j=1}^{n^{\text{te}}} H(\boldsymbol{r}_{f^\star}, \hat{\boldsymbol{\theta}}_{n^{\text{tr}}}, \boldsymbol{x}_j) \\
&\qquad - \mathbb{E}\left[ H(\boldsymbol{r}_{f^\star}, \hat{\boldsymbol{\theta}}_{n^{\text{tr}}}, \boldsymbol{x}) \right] + 2L\mathbb{E}\left[ \|\hat{\boldsymbol{\theta}}_{n^{\text{tr}}} - \boldsymbol{\theta}^\star\|_2 \right],
\end{aligned} \tag{30}$$

where in the last inequality we used the fact that $\hat{\boldsymbol{r}}_n$ is a minimizer of $\boldsymbol{r} \mapsto \frac{1}{n} \sum_{j=1}^n H(\boldsymbol{r}, \hat{\boldsymbol{\theta}}_t, \boldsymbol{x}_j)$. Finally by using Lemma E.5 and Lemma E.6 with $\delta/2$ each, we have that with probability $1 - \delta$,

$$\mathcal{L}_{\boldsymbol{\theta}^\star}(\hat{\boldsymbol{r}}_{n^{\text{te}}}) - \mathcal{L}_{\boldsymbol{\theta}^\star}(\boldsymbol{r}_{f^\star}) \leq \frac{4}{\sqrt{n^{\text{te}}}} \text{Rad}(\mathcal{F}) + 2L\mathbb{E}\left[ \|\hat{\boldsymbol{\theta}}_{n^{\text{tr}}} - \boldsymbol{\theta}^\star\|_2 \right] + 4B\sqrt{\frac{\log(4/\delta)}{n^{\text{te}}}} \tag{31}$$

Plugging this back into Equation (29), we have that

$$\|\hat{\boldsymbol{r}}_{n^{\text{te}}} - \boldsymbol{r}_{f^\star}\|_2^2 \leq \frac{2}{\mu p_{\min}} \left( \frac{4}{\sqrt{n^{\text{te}}}} \text{Rad}(\mathcal{F}) + 4B\sqrt{\frac{\log(4/\delta)}{n^{\text{te}}}} \right) + \frac{4L}{\mu p_{\min}} \mathbb{E}\left[ \|\hat{\boldsymbol{\theta}}_{n^{\text{tr}}} - \boldsymbol{\theta}^\star\|_2 \right]. \tag{32}$$

$\square$

**Lemma E.1.** *For any $\boldsymbol{r} \in \mathbb{R}_+^m$, $\boldsymbol{\theta} \in \Theta$, $\boldsymbol{x} \in \mathcal{X}$, we have that*

$$\boldsymbol{r}^\top f(\boldsymbol{x}, \boldsymbol{\theta}) \leq \frac{1}{p_{min}}.$$

*Proof.* Applying Hölder's inequality we have that

$$\boldsymbol{r}^\top f(\boldsymbol{x}, \boldsymbol{\theta}) \leq \|\boldsymbol{r}\|_\infty \|f(\boldsymbol{x}, \boldsymbol{\theta})\|_1 = \|\boldsymbol{r}\|_\infty.$$

Moreover, since $\boldsymbol{r} \in \mathbb{R}_+^m$, we have that $\sum_y r_y p_{tr}(y) = 1$ This implies that $\|\boldsymbol{r}\|_\infty \leq \frac{1}{p_{\min}}$, which yields the result. $\square$

**Lemma E.2** (Implication of Assumption Assumption 5.1). *Under Assumption 5.1, there exists $B > 0$ such that for any $\boldsymbol{r} \in \mathbb{R}_+^m$, $\boldsymbol{\theta} \in \Theta$, $\boldsymbol{x} \in \mathcal{X}$,*

$$|\log(\boldsymbol{r}^\top f(\boldsymbol{x}, \boldsymbol{\theta}))| \leq B.$$

*Proof.* Since $\boldsymbol{r} \in \mathbb{R}_+^m$, it has at least one non-zero coordinate and $f(\boldsymbol{x}, \boldsymbol{\theta})$ is the output of a softmax layer so all of its coordinates are non-zero. Consequently,

$$\boldsymbol{r}^\top f(\boldsymbol{x}, \boldsymbol{\theta}) > 0$$

So by Assumption 5.1, the function $(\boldsymbol{r}, \boldsymbol{\theta}, \boldsymbol{x}) \mapsto \log(\boldsymbol{r}^\top f(\boldsymbol{x}, \boldsymbol{\theta}))$ is defined and continuous over a compact set, so there exists a constant $B$ giving us the result. $\square$

**Lemma E.3** (Population Strong Convexity). *Let $H(\boldsymbol{r}, \boldsymbol{\theta}, \boldsymbol{x}) = -\log(\boldsymbol{r}^\top f(\boldsymbol{x}, \boldsymbol{\theta}))$. Under Assumption Assumption 5.2, the function*

$$\mathcal{L}_{\boldsymbol{\theta}^\star} : \boldsymbol{r} \mapsto \mathbb{E}\left[H(\boldsymbol{r}, \boldsymbol{\theta}^\star, \boldsymbol{x})\right]$$

*is $\mu p_{\min}$-strongly convex.*

*Proof.* We first compute the Hessian of $\mathcal{L}$ to find that

$$\nabla^2 \mathcal{L}(\boldsymbol{r}) = \mathbb{E}\left[\frac{1}{(\boldsymbol{r}^\top f(\boldsymbol{x}, \boldsymbol{\theta}^\star))^2} f(\boldsymbol{x}, \boldsymbol{\theta}^\star) f(\boldsymbol{x}, \boldsymbol{\theta}^\star)^\top\right].$$

Since by Lemma E.1, we have that $\boldsymbol{r}^\top f(\boldsymbol{x}, \boldsymbol{\theta}^\star) \leq p_{\min}^{-1}$, we conclude that

$$\nabla^2 \mathcal{L}(\boldsymbol{r}) \succeq p_{\min} \mathbb{E}\left[f(\boldsymbol{x}, \boldsymbol{\theta}^\star) f(\boldsymbol{x}, \boldsymbol{\theta}^\star)^\top\right] \succeq \mu p_{\min} \mathbf{I}_m.$$

$\square$

**Lemma E.4** (Lipschitz Parametrization). *Let $H(\boldsymbol{r}, \boldsymbol{\theta}, \boldsymbol{x}) = -\log(f(\boldsymbol{x}, \boldsymbol{\theta})^\top \boldsymbol{r})$. There exists $L > 0$ such that for any $\boldsymbol{\theta}_1, \boldsymbol{\theta}_2 \in \Theta$, and $\boldsymbol{r} \in \mathbb{R}_+^m$, we have that*
$$|H(\boldsymbol{r}, \boldsymbol{\theta}_1, \boldsymbol{x}) - H(\boldsymbol{r}, \boldsymbol{\theta}_2, \boldsymbol{x})| \leq L\|\boldsymbol{\theta}_1 - \boldsymbol{\theta}_2\|_2.$$

*Proof.* The gradient of $H$ with respect to $\boldsymbol{\theta}$ is given by

$$\nabla_{\boldsymbol{\theta}} H(\boldsymbol{r}, \boldsymbol{\theta}, \boldsymbol{x}) = -\frac{1}{f(\boldsymbol{x}, \boldsymbol{\theta})^\top \boldsymbol{r}} \nabla_{\boldsymbol{\theta}} f(\boldsymbol{x}, \boldsymbol{\theta})$$

Reasoning like in Lemma E.1, we know that $\frac{1}{f(\boldsymbol{x}, \boldsymbol{\theta})^\top \boldsymbol{r}}$ is defined and continuous over the compact set of its parameters, we also know that $f$ is a neural network parametrized by $\boldsymbol{\theta}$, hence $\nabla_{\boldsymbol{\theta}} f(\boldsymbol{x}, \boldsymbol{\theta})$ is bounded when $\boldsymbol{\theta}$ and $\boldsymbol{x}$ are bounded. Consequently, under Assumption 5.1, there exists a constant $L > 0$ such that

$$\|\nabla_{\boldsymbol{\theta}} H(\boldsymbol{r}, \boldsymbol{\theta}, \boldsymbol{x})\|_2 \leq L.$$

$\square$

**Lemma E.5** (Uniform Bound 1). *Let $\delta \in (0, 1)$, with probability $1 - \delta$, we have that*

$$\mathbb{E}\left[H(\hat{\boldsymbol{r}}_n, \hat{\boldsymbol{\theta}}_t, \boldsymbol{x})\right] - \frac{1}{n} \sum_{j=1}^n H(\hat{\boldsymbol{r}}_n, \hat{\boldsymbol{\theta}}_t, \boldsymbol{x}_j) \tag{33}$$

$$\leq \frac{2}{\sqrt{n}} Rad(\mathcal{F}) + 2B\sqrt{\frac{\log(4/\delta)}{n}}.$$

*Proof.* Let $\delta \in (0, 1)$. Since $\hat{\boldsymbol{r}}_n$ is learned from the samples $\boldsymbol{x}_j$, we do not have independence, which would have allowed us to apply a concentration inequality. Hence, we derive a uniform bound as follows. We begin by observing that:

$$\mathbb{E}\left[H(\hat{\boldsymbol{r}}_n, \hat{\boldsymbol{\theta}}_t, \boldsymbol{x})\right] - \frac{1}{n} \sum_{j=1}^n H(\hat{\boldsymbol{r}}_n, \hat{\boldsymbol{\theta}}_t, \boldsymbol{x}_j)$$

$$\leq \sup_{\boldsymbol{r}, \boldsymbol{\theta}} \left(\mathbb{E}\left[H(\boldsymbol{r}, \boldsymbol{\theta}, \boldsymbol{x})\right] - \frac{1}{n} \sum_{j=1}^n H(\boldsymbol{r}, \boldsymbol{\theta}, \boldsymbol{x}_j)\right)$$

Now since Lemma E.2 holds, we can apply McDiarmid's Inequality to get that with probability $1 - \delta$, we have:

$$\sup_{\boldsymbol{r}, \boldsymbol{\theta}} \left(\mathbb{E}\left[H(\boldsymbol{r}, \boldsymbol{\theta}, \boldsymbol{x})\right] - \frac{1}{n} \sum_{j=1}^n H(\boldsymbol{r}, \boldsymbol{\theta}, \boldsymbol{x}_j)\right)$$

$$\leq \mathbb{E}\left[\sup_{\boldsymbol{r}, \boldsymbol{\theta}} \left(\mathbb{E}[H(\boldsymbol{r}, \boldsymbol{\theta}, \boldsymbol{x})] - \frac{1}{n} \sum_{j=1}^n H(\boldsymbol{r}, \boldsymbol{\theta}, \boldsymbol{x}_j)\right)\right] + 2B\sqrt{\frac{\log(2/\delta)}{n}}$$

The expectation of the supremum on the right-hand side can be bounded by the Rademacher complexity of $\mathcal{F} := \{\boldsymbol{x} \mapsto \boldsymbol{r}^\top f(\boldsymbol{x}, \boldsymbol{\theta}), \ (\boldsymbol{r}, \boldsymbol{\theta}) \in \mathbb{R}_+^m \times \Theta\}$, and we obtain:

$$
\sup_{\boldsymbol{r}, \boldsymbol{\theta}} \left( \mathbb{E}\big[H(\boldsymbol{r}, \boldsymbol{\theta}, \boldsymbol{x})\big] - \frac{1}{n} \sum_{j=1}^n H(\boldsymbol{r}, \boldsymbol{\theta}, \boldsymbol{x}_j) \right)
$$
$$
\leq \frac{2}{\sqrt{n}} \text{Rad}(\mathcal{F}) + 2B \sqrt{\frac{\log(2/\delta)}{n}}. \tag{34}
$$

$\square$

**Lemma E.6** (Uniform Bound 2). *Let $\delta \in (0,1)$, with probability $1 - \delta$, we have that*

$$
\mathbb{E}\left[H(\boldsymbol{r}_{f^\star}, \hat{\boldsymbol{\theta}}_t, \boldsymbol{x})\right] - \frac{1}{n} \sum_{j=1}^n H(\boldsymbol{r}_{f^\star}, \hat{\boldsymbol{\theta}}_t, \boldsymbol{x}_j)
$$
$$
\leq \frac{2}{\sqrt{n}} Rad(\mathcal{F}) + 2B \sqrt{\frac{\log(2/\delta)}{n}}. \tag{35}
$$

*Proof.* The proof is identical to that of Lemma E.5. $\square$

**Lemma E.7** (Strong Convexity of Population Loss). *Let $\mathcal{L}(\boldsymbol{r}, \boldsymbol{\theta})$ be the population loss as defined in Lemma E.7. We establish that $\mathcal{L}(\boldsymbol{r}, \boldsymbol{\theta})$ is $\mu p_{\min}$-strongly convex under the assumptions of calibration (Assumption 5.2).*

*Proof.* We compute the Hessian of the population loss $\mathcal{L}$ as in Lemma E.7, obtaining that:

$$
\nabla^2 \mathcal{L}(\boldsymbol{r}) = \mathbb{E}\left[\frac{1}{(\boldsymbol{r}^\top f(\boldsymbol{x}, \boldsymbol{\theta}))^2} f(\boldsymbol{x}, \boldsymbol{\theta}) f(\boldsymbol{x}, \boldsymbol{\theta})^\top\right].
$$

From Lemma E.1, we have that $\boldsymbol{r}^\top f(\boldsymbol{x}, \boldsymbol{\theta}) \leq p_{\min}^{-1}$. Therefore, we conclude:

$$
\nabla^2 \mathcal{L}(\boldsymbol{r}) \succeq p_{\min} \mathbb{E}\left[f(\boldsymbol{x}, \boldsymbol{\theta}) f(\boldsymbol{x}, \boldsymbol{\theta})^\top\right] \succeq \mu p_{\min} \mathbf{I}_m.
$$

$\square$

**Lemma E.8** (Bound on Empirical Loss). *Under Assumption 5.1, the empirical loss $\mathcal{L}_{n^{te}}(\boldsymbol{r}, \hat{\boldsymbol{\theta}}_{n^{tr}})$ satisfies the following concentration bound:*

$$
\mathbb{P}\left( \sup_{\boldsymbol{r} \in \mathbb{R}_+^m} \left| \mathcal{L}_{n^{te}}(\boldsymbol{r}, \hat{\boldsymbol{\theta}}_{n^{tr}}) - \mathcal{L}(\boldsymbol{r}, \hat{\boldsymbol{\theta}}_{n^{tr}}) \right| > \epsilon \right) \leq 2 \exp\left(-c n^{te} \epsilon^2\right).
$$

*Proof.* This result follows from standard concentration inequalities, such as McDiarmid's inequality, together with the Lipschitz continuity of the loss function $\mathcal{L}$ with respect to the samples. $\square$

## F   PROOF OF THEOREM 5.2 AND CONVERGENCE-COMMUNICATION GUARANTEES FOR IW-ERM WITH VRLS

We now establish convergence rates for IW-ERM with VRLS and show our proposed importance weighting achieves *the same rates* with the data-dependent *constant terms* increase linearly with $\max_{y \in \mathcal{Y}} \sup_f r_f(y) = r_{\max}$ under negligible communication overhead over the baseline ERM-solvers without importance weighting. In Appendix F, we establish tight convergence rates and communication guarantees for IW-ERM with VRLS in a broad range of importance optimization settings including convex optimization, second-order differentiability, composite optimization with proximal operator, optimization with adaptive step-sizes, and nonconvex optimization, along the lines of e.g., (Woodworth et al., 2020; Haddadpour et al., 2021; Glasgow et al., 2022; Liu et al., 2023; Hu & Huang, 2023; Wu et al., 2023; Liu et al., 2023).

By estimating the ratios locally and absorbing into local losses, we note that the properties of the modified local loss w.r.t. the neural network parameters $\boldsymbol{w}$, e.g., convexity and smoothness, do not change. The data-dependent parameters such as Lipschitz and smoothness constants for $\ell \circ h_{\boldsymbol{w}}$ w.r.t. $\boldsymbol{w}$ are scaled linearly by $r_{\max}$. Our method of importance ratio estimation trains the pre-defined predictor *exclusively using local training data*, which implies IW-ERM with VRLS achieves the same privacy guarantees as the baseline ERM-solvers without importance weighting. For ratio estimation, the communication between clients involves only the estimated marginal label distribution, instead of data, ensuring negligible communication overhead. Given the size of variables to represent marginal distributions, which is by orders of magnitude smaller than the number of parameters of the underlying neural networks for training and the fact that ratio estimation involves only one round of communication, the overall communication overhead for ratio estimation is masked by the communication costs of model training. The communication costs for IW-ERM with VRLS over the course of optimization are exactly the same as those of the baseline ERM-solvers without importance weighting. All in all, importance weighting does not negatively impact communication guarantees throughout the course of optimization, which proves Theorem 5.2.

In the following, we establish tight convergence rates and communication guarantees for IW-ERM with VRLS in a broad range of importance optimization settings including convex optimization, second-order differentiability, composite optimization with proximal operator, optimization with adaptive step-sizes, and nonconvex optimization.

For convex and second-order Differentiable optimization, we establish a lower bound on the convergence rates for IW-ERM in with VRLS and local updating along the lines of e.g., (Glasgow et al., 2022, Theorem 3.1).

**Assumption F.1** (PL with Compression). *1) The $\ell(h_{\boldsymbol{w}}(\boldsymbol{x}), y)$ is $\beta$-smoothness and convex w.r.t. $\boldsymbol{w}$ for any $(\boldsymbol{x}, y)$ and satisfies Polyak-Łojasiewicz (PL) condition (there exists $\alpha_\ell > 0$ such that, for all $\boldsymbol{w} \in \mathcal{W}$, we have $\ell(h_{\boldsymbol{w}}) \leq \|\nabla_{\boldsymbol{w}} \ell(h_{\boldsymbol{w}})\|_2^2 / (2\alpha_\ell)$); 2) The compression scheme $\mathcal{Q}$ is unbiased with bounded variance, i.e., $\mathbb{E}[\mathcal{Q}(\boldsymbol{x})] = \boldsymbol{x}$ and $\mathbb{E}[\|\mathcal{Q}(\boldsymbol{x}) - \boldsymbol{x}\|_2^2 \leq q\|\boldsymbol{x}\|_2^2]$; 3) The stochastic gradient $\boldsymbol{g}(\boldsymbol{w}) = \widetilde{\nabla}_{\boldsymbol{w}} \ell(h_{\boldsymbol{w}})$ is unbiased, i.e., $\mathbb{E}[\boldsymbol{g}(\boldsymbol{w})] = \nabla_{\boldsymbol{w}} \ell(h_{\boldsymbol{w}})$ for any $\boldsymbol{w} \in \mathcal{W}$ with bounded variance $\mathbb{E}[\|\boldsymbol{g}(\boldsymbol{w}) - \nabla_{\boldsymbol{w}} \ell(h_{\boldsymbol{w}})\|_2^2]$.*

For nonconvex optimization with PL condition and communication compression, we establish convergence and communication guarantees for IW-ERM with VRLS, compression, and local updating along the lines of e.g., (Haddadpour et al., 2021, Theorem 5.1).

**Theorem F.1** (Convergence and Communication Bounds for Nonconvex Optimization with PL). *Let $\kappa$ denote the condition number, $\tau$ denote the number of local steps, $R$ denote the number of communication rounds, and $\max_{y \in \mathcal{Y}} \sup_f r_f(y) = r_{\max}$. Under Assumption F.1, suppose Algorithm 2 with $\tau$ local updates and communication compression (Haddadpour et al., 2021, Algorithm 1) is run for $T = \tau R$ total stochastic gradients per node with fixed step-sizes $\eta = 1/(2r_{\max}\beta\gamma\tau(q/K + 1))$ and $\gamma \geq K$. Then we have $\mathbb{E}[\ell(h_{\boldsymbol{w}_T}) - \ell(h_{\boldsymbol{w}^\star})] \leq \epsilon$ by setting*

$$R \lesssim \left(\frac{q}{K} + 1\right)\kappa \log\left(\frac{1}{\epsilon}\right) \quad and \quad \tau \lesssim \left(\frac{q + 1}{K(q/K + 1)\epsilon}\right). \tag{36}$$

**Assumption F.2** (Nonconvex Optimization with Adaptive Step-sizes). *1) The $\ell \circ h_{\boldsymbol{w}}$ is $\beta$-smoothness with bounded gradients; 2) The stochastic gradients $\boldsymbol{g}(\boldsymbol{w}) = \widetilde{\nabla}_{\boldsymbol{w}} \ell(h_{\boldsymbol{w}})$ is unbiased with bounded variance $\mathbb{E}[\|\boldsymbol{g}(\boldsymbol{w}) - \nabla_{\boldsymbol{w}} \ell(h_{\boldsymbol{w}})\|_2^2]$; 3) Adaptive matrices $A_t$ constructed as in (Wu et al., 2023, Algorithm 2) are diagonal and the minimum eigenvalues satisfy $\lambda_{\min}(A_t) \geq \rho > 0$ for some $\rho \in \mathbb{R}_+$.*

For nonconvex optimization with adaptive step-sizes, we establish convergence and communication guarantees for IW-ERM with VRLS and local updating along the lines of e.g., (Wu et al., 2023, Theorem 2).

**Theorem F.2** (Convergence and Communication Guarantees for Nonconvex Optimization with Adaptive Step-sizes). *Let $\tau$ denote the number of local steps, $R$ denote the number of communication rounds, and $\max_{y \in \mathcal{Y}} \sup_f r_f(y) = r_{\max}$. Under Assumption F.2, suppose Algorithm 2 with $\tau$ local updates is run for $T = \tau R$ total stochastic gradients per node with an adaptive step-size similar to (Wu et al., 2023, Algorithm 2). Then we $\mathbb{E}[\|\nabla_{\boldsymbol{w}} \ell(h_{\boldsymbol{w}_T})\|_2] \leq \epsilon$ by setting:*

$$T \lesssim \frac{r_{\max}}{K\epsilon^3} \quad and \quad R \lesssim \frac{r_{\max}}{\epsilon^2}. \tag{37}$$

**Assumption F.3** (Composite Optimization with Proximal Operator). *1) The $\ell \circ h_{\boldsymbol{w}}$ is smooth and strongly convex with condition number $\kappa$; 2) The stochastic gradients $\boldsymbol{g}(\boldsymbol{w}) = \widetilde{\nabla}_{\boldsymbol{w}} \ell(h_{\boldsymbol{w}})$ is unbiased.*

For composite optimization with strongly convex and smooth functions and proximal operator, we establish an upper bound on oracle complexity to achieve $\epsilon$ error on the Lyapunov function defined as in (Hu & Huang, 2023, Section 4) for Gradient Flow-type transformation of IW-ERM with VRLS in the limit of infinitesimal step-size.

**Theorem F.3** (Oracle Complexity of Proximal Operator for Composite Optimization). *Let $\kappa$ denote the condition number. Under Assumption F.3, suppose Gradient Flow-type transformation of Algorithm 2 with VRLS and Proximal Operator evolves in the limit of infinitesimal step-size (Hu & Huang, 2023, Algorithm 3). Then it achieves $\mathcal{O}\left(r_{\max}\sqrt{\kappa}\log(1/\epsilon)\right)$ Proximal Operator Complexity.*

## G  COMPLEXITY ANALYSIS

In our algorithm, the ratio estimation is performed once in parallel before the IW-ERM step.

In the experiments, we used a simple network to estimate the ratios in advance, which required significantly less computational effort compared to training the global model. Although IW-ERM with VRLS introduces additional computational complexity compared to the baseline FedAvg, it results in substantial improvements in overall generalization, particularly under challenging label shift conditions.

# H   MATHEMATICAL NOTATIONS

In this appendix, we provide a summary of mathematical notations used in this paper in Table 5:

Table 5: Math Symbols

| Math Symbol | Definition |
|---|---|
| $\mathcal{X}$ | Compact metric space for features |
| $\mathcal{Y}$ | Discrete label space with $|\mathcal{Y}| = m$ |
| $K$ | Number of clients in an FL setting |
| $\mathcal{S}_k$ | All samples in the training set of client $k$ |
| $h_{\boldsymbol{w}}$ | Hypothesis function $h_{\boldsymbol{w}} : \mathcal{X} \to \mathcal{Y}$ |
| $\mathcal{H}$ | Hypothesis class for $h_{\boldsymbol{w}}$ |
| $\mathcal{Z}$ | Mapping space from $\mathcal{X}$, which can be discrete or continuous |

## I  LIMITATIONS

The distribution shifts observed in real-world data are often not fully captured by the label shift or relaxed distribution shift assumptions. In our experiments, we applied mild test data augmentation to approximate the relaxed label shift and manage ratio estimation errors for both the baselines and our method. However, the label shift assumption remains overly restrictive, and the relaxed label shift lacks robust empirical validation in practical scenarios.

Additionally, IW-ERM's parameter estimation relies on local predictors at each client, which limits its scalability. In practice, a simpler global predictor could be sufficient for parameter estimation and IW-ERM training. Future research could explore VRLS variants capable of effectively handling more complex distribution shifts in challenging datasets, such as CIFAR-10.1 (Recht et al., 2018; Torralba et al., 2008), as suggested in (Garg et al., 2023).

## J EXPERIMENTAL DETAILS AND ADDITIONAL EXPERIMENTS

In this section, we provide experimental details and additional experiments. In particular, we validate our theory on multiple clients in a federated setting and show that our IW-ERM outperforms FedAvg and FedBN baselines *under drastic and challenging label shifts*.

### J.1 EXPERIMENTAL DETAILS

In single-client experiments, a simple MLP without dropout is used as the predictor for MNIST, and ResNet-18 for CIFAR-10.

For experiments in a federated learning setting, both MNIST (LeCun et al., 1998) and Fashion MNIST (Xiao et al., 2017) datasets are employed, each containing 60,000 training samples and 10,000 test samples, with each sample being a 28 by 28 pixel grayscale image. The CIFAR-10 dataset (Krizhevsky) comprises 60,000 colored images, sized 32 by 32 pixels, spread across 10 classes with 6,000 images per class; it is divided into 50,000 training images and 10,000 test images. In this setting, the objective is to minimize the cross-entropy loss. Stochastic gradients for each client are calculated with a batch size of 64 and aggregated on the server using the Adam optimizer. LeNet is used for experiments on MNIST and Fashion MNIST with a learning rate of 0.001 and a weight decay of $1 \times 10^{-6}$. For CIFAR-10, ResNet-18 is employed with a learning rate of 0.0001 and a weight decay of 0.0001. Three independent runs are implemented for 5-client experiments on Fashion MNIST and CIFAR-10, while for 10 clients, one run is conducted on CIFAR-10. The regularization coefficient $\zeta$ in Equation (4) is set to 1 for all experiments. All experiments are performed using a single GPU on an internal cluster and Colab.

Importantly, the training of the predictor for ratio estimation on both the baseline MLLS and our VRLS is executed with identical hyperparameters and epochs for CIFAR-10 and Fashion MNIST. The training is halted once the classification loss reaches a predefined threshold on MNIST.

### J.2 RELAXED LABEL SHIFT EXPERIMENTS

In conventional label shift, it is assumed that $p(\boldsymbol{x} \mid y)$ remains unchanged across training and test data. However, this assumption is often too strong for real-world applications, such as in healthcare, where different hospitals may use varying equipment, leading to shifts in $p(\boldsymbol{x} \mid y)$ even with the same labels (Rajendran et al., 2023). Relaxed label shift loosens this assumption by allowing small changes in the conditional distribution (Garg et al., 2023; Luo & Ren, 2022).

To formalize this, we use the distributional distance $\mathcal{D}$ and a relaxation parameter $\epsilon > 0$, as defined by Garg et al. (2023): $\max_y \mathcal{D}\left(p_{\text{tr}}(\boldsymbol{x} \mid y), p_{\text{te}}(\boldsymbol{x} \mid y)\right) \leq \epsilon$. This allows for slight differences in feature distributions between training and testing, capturing a more realistic scenario where the conditional distribution is not strictly invariant.

In our case, visual inspection suggests that the differences between temporally distinct datasets, such as CIFAR-10 and CIFAR-10.1_v6 (Torralba et al., 2008; Recht et al., 2018), may not meet the assumption of a small $\epsilon$. To address this, we instead simulate controlled shifts using test data augmentation, allowing us to regulate the degree of relaxation, following the approach outlined in Garg et al. (2023).

### J.3 ADDITIONAL EXPERIMENTS

In this section, we provide supplementary results, visualizations of accuracy across clients and tables showing dataset distribution in FL setting and relaxed label shift.

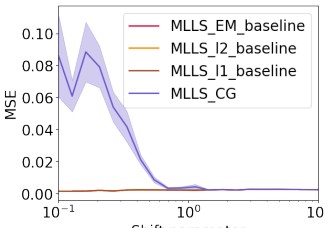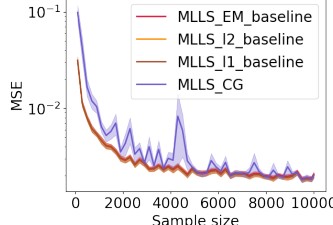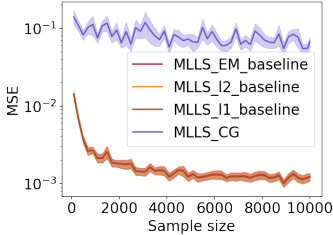

Figure 3: MSE analysis on MNIST for MLLS baselines. **Left:** Performance evaluation across various alpha values, comparing different methods: MLLS_EM, MLLS_L1, MLLS_L2, and MLLS_CG. MLLS_L1 and MLLS_L2 utilize convex optimization with $L_1$ and $L_2$ regularization for estimating our limited test sample problem, respectively, and are solved directly with a convex solver. In contrast, MLLS_CG uses conjugate gradient descent and MLLS_EM solves this convex optimization problem with EM algorithm. Both the EM and convex optimization methods (MLLS_L1, MLLS_L2) demonstrate superior and more consistent performance, especially under severe label shift conditions, when compared to MLLS_CG. **Middle:** At an alpha value of 1.0, the MSE analysis shows comparable performance across most methods, with the exception of MLLS_CG, which lags behind. **Right:** For alpha=0.1, MLLS_CG performs significantly worse than the EM and convex optimization methods, consistent with the trends observed in the left plot.

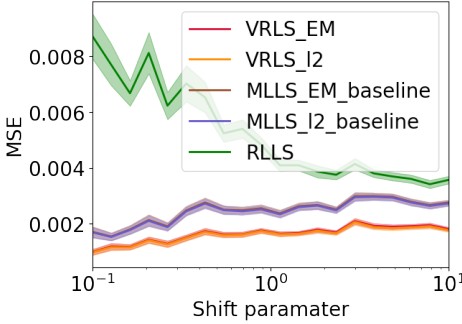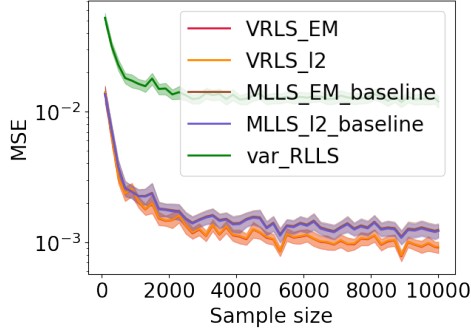

Figure 4: In our detailed analysis with the MNIST dataset, we conduct a thorough comparison of VRLS alongside MLLS (Garg et al., 2020), EM (Saerens et al., 2002), and also RLLS (Azizzade-nesheli et al., 2019).

Table 6: LeNet on Fashion MNIST with label shift across 5 clients. 15,000 iterations for FedAvg and FedBN; 5,000 for Upper Bound (FTW-ERM) using true ratios and our IW-ERM. To mention, to train our predictor, we use a simpliest MLP and employ linear kernel.

| FMNIST | Our IW-ERM | FedAvg | FedBN | Upper Bound |
|---|---|---|---|---|
| **Avg. accuracy** | $\mathbf{0.7520 \pm 0.0209}$ | $0.5472 \pm 0.0297$ | $0.5359 \pm 0.0306$ | $0.8273 \pm 0.0041$ |
| Client 1 accuracy | $\mathbf{0.7162 \pm 0.0059}$ | $0.3616 \pm 0.0527$ | $0.3261 \pm 0.0296$ | $0.8590 \pm 0.0062$ |
| Client 2 accuracy | $\mathbf{0.9266 \pm 0.0125}$ | $0.9060 \pm 0.0157$ | $0.9035 \pm 0.0162$ | $0.9357 \pm 0.0037$ |
| Client 3 accuracy | $\mathbf{0.6724 \pm 0.0467}$ | $0.3279 \pm 0.0353$ | $0.3612 \pm 0.0814$ | $0.7896 \pm 0.0109$ |
| Client 4 accuracy | $\mathbf{0.7979 \pm 0.0448}$ | $0.6858 \pm 0.0105$ | $0.6654 \pm 0.0121$ | $0.8098 \pm 0.0112$ |
| Client 5 accuracy | $\mathbf{0.6468 \pm 0.0248}$ | $0.4548 \pm 0.0655$ | $0.4234 \pm 0.0387$ | $0.7426 \pm 0.0257$ |

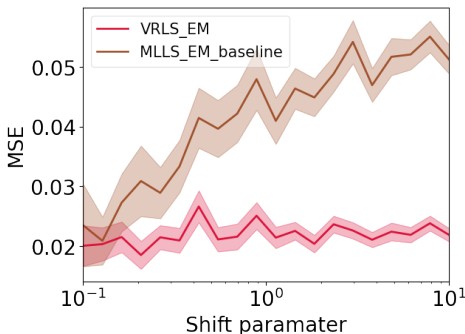 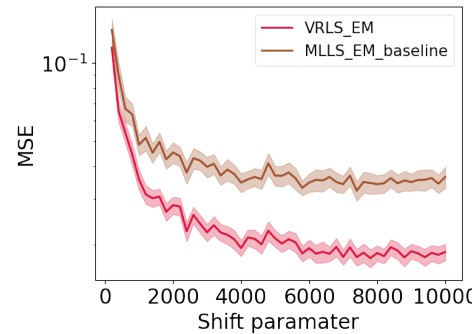

Figure 5: In this experiment with Fashion MNIST, a simple MLP with dropout were employed.

Table 7: ResNet-18 on CIFAR-10 with label shift across 5 clients. For fair comparison, we run 5,000 iterations for our method and Upper Bound, while 10000 for FedAvg and FedBN.

| CIFAR-10 | Our IW-ERM | FedAvg | FedBN | Upper Bound |
|---|---|---|---|---|
| **Avg. accuracy** | **$0.5640 \pm 0.0241$** | $0.4515 \pm 0.0148$ | $0.4263 \pm 0.0975$ | $0.5790 \pm 0.0103$ |
| Client 1 accuracy | **$0.6410 \pm 0.0924$** | $0.5405 \pm 0.1845$ | $0.5321 \pm 0.0620$ | $0.7462 \pm 0.0339$ |
| Client 2 accuracy | **$0.8434 \pm 0.0359$** | $0.3753 \pm 0.0828$ | $0.4656 \pm 0.2158$ | $0.7509 \pm 0.0534$ |
| Client 3 accuracy | **$0.4591 \pm 0.1131$** | $0.3973 \pm 0.1333$ | $0.2838 \pm 0.1055$ | $0.5845 \pm 0.0854$ |
| Client 4 accuracy | **$0.4751 \pm 0.1241$** | $0.5007 \pm 0.1303$ | $0.5256 \pm 0.1932$ | $0.3507 \pm 0.0578$ |
| Client 5 accuracy | **$0.4013 \pm 0.0430$** | $0.4429 \pm 0.1195$ | $0.5603 \pm 0.1581$ | $0.4627 \pm 0.0456$ |

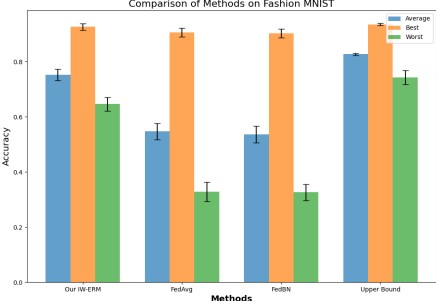

Figure 6: The average, best-client, and worst-client accuracy, along with their standard deviations, are derived from Table 6. Our method exhibits the lowest standard deviation, showcasing the most robust accuracy amongst the compared methods.

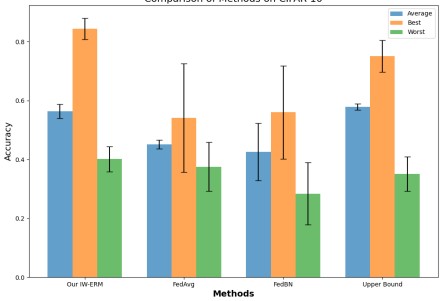

Figure 7: The average, best-client, and worst-client accuracy, along with their standard deviations, are derived from Table 7.

Table 8: Label distribution on Fasion MNIST with 5 nodes, with the majority of classes possessing a limited number of training and test images across each node.

| | | Class | | | | | | | | | |
|---|---|---|---|---|---|---|---|---|---|---|---|
| | | 0 | 1 | 2 | 3 | 4 | 5 | 6 | 7 | 8 | 9 |
| Node 1 | Train | 34 | 34 | 34 | 34 | 34 | 5862 | 34 | 34 | 34 | 34 |
| | Test | 977 | 5 | 5 | 5 | 5 | 5 | 5 | 5 | 5 | 5 |
| Node 2 | Train | 34 | 34 | 34 | 34 | 34 | 34 | 5862 | 34 | 34 | 34 |
| | Test | 5 | 977 | 5 | 5 | 5 | 5 | 5 | 5 | 5 | 5 |
| Node 3 | Train | 34 | 34 | 34 | 34 | 34 | 34 | 34 | 5862 | 34 | 34 |
| | Test | 5 | 5 | 977 | 5 | 5 | 5 | 5 | 5 | 5 | 5 |
| Node 4 | Train | 34 | 34 | 34 | 34 | 34 | 34 | 34 | 34 | 5862 | 34 |
| | Test | 5 | 5 | 5 | 977 | 5 | 5 | 5 | 5 | 5 | 5 |
| Node 5 | Train | 34 | 34 | 34 | 34 | 34 | 34 | 34 | 34 | 34 | 5862 |
| | Test | 5 | 5 | 5 | 5 | 977 | 5 | 5 | 5 | 5 | 5 |

Table 9: Label distribution on CIFAR-10 with 5 clients, with the majority of classes possessing a limited number of training and test images across each client.

| | | Class | | | | | | | | | |
|---|---|---|---|---|---|---|---|---|---|---|---|
| | | 0 | 1 | 2 | 3 | 4 | 5 | 6 | 7 | 8 | 9 |
| Node 1 | Train | 34 | 34 | 34 | 34 | 34 | 5862 | 34 | 34 | 34 | 34 |
| | Test | 977 | 5 | 5 | 5 | 5 | 5 | 5 | 5 | 5 | 5 |
| Node 2 | Train | 34 | 34 | 34 | 34 | 34 | 34 | 5862 | 34 | 34 | 34 |
| | Test | 5 | 977 | 5 | 5 | 5 | 5 | 5 | 5 | 5 | 5 |
| Node 3 | Train | 34 | 34 | 34 | 34 | 34 | 34 | 34 | 5862 | 34 | 34 |
| | Test | 5 | 5 | 977 | 5 | 5 | 5 | 5 | 5 | 5 | 5 |
| Node 4 | Train | 34 | 34 | 34 | 34 | 34 | 34 | 34 | 34 | 5862 | 34 |
| | Test | 5 | 5 | 5 | 977 | 5 | 5 | 5 | 5 | 5 | 5 |
| Node 5 | Train | 34 | 34 | 34 | 34 | 34 | 34 | 34 | 34 | 34 | 5862 |
| | Test | 5 | 5 | 5 | 5 | 977 | 5 | 5 | 5 | 5 | 5 |

Table 10: Label distribution on CIFAR-10 with 100 clients, wherein groups of 10 clients share the same distribution and ratios. The majority of classes possess a limited quantity of training and test images on each client.

| | | Class | | | | |
| --- | --- | --- | --- | --- | --- | --- |
| | | 0 | 1 | 2 | 3 | 4 |
| Client 1-10 | Train | $95/100$ | $5/9$ | $5/9$ | $5/9$ | $5/9$ |
| | Test | $5/9$ | $5/9$ | $5/9$ | $5/9$ | $5/9$ |
| Client 11-20 | Train | $5/9$ | $95/100$ | $5/9$ | $5/9$ | $5/9$ |
| | Test | $5/9$ | $5/9$ | $5/9$ | $5/9$ | $5/9$ |
| Client 21-30 | Train | $5/9$ | $5/9$ | $95/100$ | $5/9$ | $5/9$ |
| | Test | $5/9$ | $5/9$ | $5/9$ | $5/9$ | $5/9$ |
| Client 31-40 | Train | $5/9$ | $5/9$ | $5/9$ | $95/100$ | $5/9$ |
| | Test | $5/9$ | $5/9$ | $5/9$ | $5/9$ | $5/9$ |
| Client 41-50 | Train | $5/9$ | $5/9$ | $5/9$ | $5/9$ | $95/100$ |
| | Test | $5/9$ | $5/9$ | $5/9$ | $5/9$ | $5/9$ |
| Client 51-60 | Train | $5/9$ | $5/9$ | $5/9$ | $5/9$ | $5/9$ |
| | Test | $5/9$ | $5/9$ | $5/9$ | $5/9$ | $95/100$ |
| Client 61-70 | Train | $5/9$ | $5/9$ | $5/9$ | $5/9$ | $5/9$ |
| | Test | $5/9$ | $5/9$ | $5/9$ | $95/100$ | $5/9$ |
| Client 71-80 | Train | $5/9$ | $5/9$ | $5/9$ | $5/9$ | $5/9$ |
| | Test | $5/9$ | $5/9$ | $95/100$ | $5/9$ | $5/9$ |
| Client 81-90 | Train | $5/9$ | $5/9$ | $5/9$ | $5/9$ | $5/9$ |
| | Test | $5/9$ | $95/100$ | $5/9$ | $5/9$ | $5/9$ |
| Client 91-100 | Train | $5/9$ | $5/9$ | $5/9$ | $5/9$ | $5/9$ |
| | Test | $95/100$ | $5/9$ | $5/9$ | $5/9$ | $5/9$ |

| | | Class | | | | |
| --- | --- | --- | --- | --- | --- | --- |
| | | 5 | 6 | 7 | 8 | 9 |
| Client 1-10 | Train | $5/9$ | $5/9$ | $5/9$ | $5/9$ | $5/9$ |
| | Test | $5/9$ | $5/9$ | $5/9$ | $5/9$ | $95/100$ |
| Client 11-20 | Train | $5/9$ | $5/9$ | $5/9$ | $5/9$ | $5/9$ |
| | Test | $5/9$ | $5/9$ | $5/9$ | $95/100$ | $5/9$ |
| Client 21-30 | Train | $5/9$ | $5/9$ | $5/9$ | $5/9$ | $5/9$ |
| | Test | $5/9$ | $5/9$ | $95/100$ | $5/9$ | $5/9$ |
| Client 31-40 | Train | $5/9$ | $5/9$ | $5/9$ | $5/9$ | $5/9$ |
| | Test | $5/9$ | $95/100$ | $5/9$ | $5/9$ | $5/9$ |
| Client 41-50 | Train | $5/9$ | $5/9$ | $5/9$ | $5/9$ | $5/9$ |
| | Test | $95/100$ | $5/9$ | $5/9$ | $5/9$ | $5/9$ |
| Client 51-60 | Train | $95/100$ | $5/9$ | $5/9$ | $5/9$ | $5/9$ |
| | Test | $5/9$ | $5/9$ | $5/9$ | $5/9$ | $5/9$ |
| Client 61-70 | Train | $5/9$ | $95/100$ | $5/9$ | $5/9$ | $5/9$ |
| | Test | $5/9$ | $5/9$ | $5/9$ | $5/9$ | $5/9$ |
| Client 71-80 | Train | $5/9$ | $5/9$ | $95/100$ | $5/9$ | $5/9$ |
| | Test | $5/9$ | $5/9$ | $5/9$ | $5/9$ | $5/9$ |
| Client 81-90 | Train | $5/9$ | $5/9$ | $5/9$ | $95/100$ | $5/9$ |
| | Test | $5/9$ | $5/9$ | $5/9$ | $5/9$ | $5/9$ |
| Client 91-100 | Train | $5/9$ | $5/9$ | $5/9$ | $5/9$ | $95/100$ |
| | Test | $5/9$ | $5/9$ | $5/9$ | $5/9$ | $5/9$ |

