# OpenReview forum: "Addressing Label Shift in Distributed Learning via Entropy Regularization​"
_ICLR.cc/2025/Conference — ICLR 2025 Poster_

### Official Review · Reviewer_L7cy · 2024-10-26

**Soundness:** 2
**Presentation:** 3
**Contribution:** 3
**Rating:** 6
**Confidence:** 3

**Summary:**

This paper proposes the Versatile Robust Label Shift (VRLS) method, which enhances the maximum likelihood estimation of the test-to-train label density ratio, for multi-node distributed learning. It incorporates Shannon entropy-based regularization and adjusts the density ratio during training to better handle label shifts at the test time. Experiments conducted on MNIST, Fashion MNIST, and CIFAR-10 demonstrate the effectiveness of the proposed VRLS. The main contributions of this paper are:

- It proposes VRLS to enhance the training of the predictor of the conditional distribution by incorporating a novel regularization term based on Shannon entropy.

- It establishes high-probability estimation error bounds for VRLS, as well as high-probability convergence bounds for IW-ERM with VRLS in nonconvex optimization settings.

**Strengths:**

- The problem studied in this paper is interesting.

- The paper is well-organized, which is easy to follow.

- This paper provides some novel perspectives for multi-node distributed learning.

**Weaknesses:**

- In my opinion, the concept of "underlying uncertainty" needs to be further elaborated. Specifically, why does the failure of a learned predictor to inherently capture the underlying uncertainty have an impact on model performance? What is the Role of "underlying uncertainty" in multi-node distributed learning?
- The motivation for the introduction of Shannon entropy-based regularization also should be further explained by the authors. What is the advantage of the leveraged technology compared with others?
- For the experimental evaluation, this paper compares the proposed method with some baseline methods. The experimental results seem promising. However, the comparison methods in this paper are all proposed before 2021, and lack the latest algorithms (e.g. [1]-[2]). I think the comparisons between the proposed method and the latest algorithms are necessary.



[1] [GradMA: A Gradient-Memory-based Accelerated Federated Learning with Alleviated Catastrophic Forgetting. CVPR 2023: 3708-3717.](https://ieeexplore.ieee.org/document/10203330/)

[2] [Multi-Level Branched Regularization for Federated Learning. ICML 2022: 11058-11073.](https://proceedings.mlr.press/v162/kim22a.html)

**Questions:**

1. The concept of "underlying uncertainty" needs to be further elaborated. Specifically, why does the failure of a learned predictor to inherently capture the underlying uncertainty have an impact on model performance? What is the Role of "underlying uncertainty" in multi-node distributed learning?
2. The motivation for the introduction of Shannon entropy-based regularization also should be further explained by the authors. What is the advantage of the leveraged technology compared with others?
3. The comparison methods in this paper are all proposed before 2021, and lack the latest algorithms. The comparisons between the proposed method and the latest algorithms are necessary.

---

> ### Author Response · Authors · 2024-11-21
> **Response to Reviewer L7cy**
>
> 1. **The concept of "underlying uncertainty" needs to be further elaborated. Specifically, why does the failure of a learned predictor to inherently capture the underlying uncertainty have an impact on model performance? What is the Role of "underlying uncertainty" in multi-node distributed learning?**
>
> In our context, “underlying uncertainty” refers to the predictor $f_{\theta}(x)$'s ability to accurately approximate the true conditional distribution $p_{tr}(y|x)$ [1], which is critical for estimating the label shift density ratio at each node in our distributed learning setups. When finite-sample and miscalibration errors [1] cause deviations in the predictor’s estimation of $p_{tr}(y|x)$, density ratio estimates on individual nodes become inaccurate, negatively affecting the global model’s performance under IW-ERM [3, 4].
>
> VRLS mitigates this issue by incorporating entropy-based regularization, which improves calibration and reduces miscalibration errors during training [2–8, 13]. This approach enhances density ratio estimation across nodes, supporting robust global performance. A concise version of this explanation is provided in L60–70 of the revised manuscript.
>
> 2. **The motivation for the introduction of Shannon entropy-based regularization also should be further explained by the authors. What is the advantage of the leveraged technology compared with others?**
>
> To estimate label shift density ratios, we build on MLLS [1], which attributes errors to finite-sample and miscalibration effects. Prior work [2–8, 13] improves calibration through post-hoc methods (e.g., temperature scaling [3]) and regularization approaches (e.g., label smoothing [6], focal loss [8], entropy penalties [5]), leveraging entropy maximization principles. For example, focal loss $\mathcal{L}_f$ balances KL divergence $\text{KL}(q \parallel \hat{p})$ (where $q$ is the true label distribution and $\hat{p}$ is the predicted distribution) and prediction entropy $\mathbb{H}[\hat{p}]$ using a hyperparameter $\gamma$:
>
> $$
> \mathcal{L}_f \geq \text{KL}(q \parallel \hat{p}) + \mathbb{H}[q] - \gamma \mathbb{H}[\hat{p}].
> $$
>
> Our work is the first to apply an **explicit** entropy regularizer to address miscalibration during training and improve density ratio estimation. Supported by theoretical guarantees and extensive experiments, VRLS outperforms baselines across varying label shift severities and test set sizes. This explicit entropy regularization serves as a proxy for the broader class of entropy-maximization methods, integrating their principles directly into the label shift estimation.
>
> 3. **The comparison methods in this paper are all proposed before 2021, and lack the latest algorithms. The comparisons between the proposed method and the latest algorithms are necessary.**
>
> Our response to this comment is addressed in Common Response 3. The suggested baselines, [10] and [11], employ ERM for distributed learning, addressing client heterogeneity through hybrid subnetworks and gradient memory with quadratic constraints. However, they do not account for label shift with statistical guarantees. Our IW-ERM approach, specifically designed for label shift, fills this gap.
>
>
> We compared our IW-ERM with FedAvg [12], GradMA [10], and FedMLB [11] on binary classification datasets carefully designed to reflect varying levels of classification difficulty. These datasets feature controlled adjustments in noise levels, covariance structures, and sample overlap, simulating diverse and realistic classification scenarios.
>
> Results show IW-ERM consistently outperforms FedAvg, FedMLB and GradMA across all datasets, demonstrating its statistical robustness and effectiveness under label shift.
>
>
> | **Datasets** | **Method**       | **Accuracy (%)** |
> |--------------|------------------|------------------|
> | **Dataset 1** | FedAvg           | 59.38           |
> |               | FedMLB           | 62.50           |
> |               |    GradMA     |       63.62     |
> |               | IW-ERM (Ours)    | **65.88**       |
> | **Dataset 2** | FedAvg           | 60.50           |
> |               | FedMLB           | 62.25           |
> |               |    GradMA     |      62.38     |
> |               | IW-ERM (Ours)    | **63.50**       |
> | **Dataset 3** | FedAvg           | 62.50           |
> |               | FedMLB           | 64.00           |
> |               |   GradMA      |      64.62      |
> |               | IW-ERM (Ours)    | **65.63**       |

---

> ### Author Response · Authors · 2024-11-21
> **Response to Reviewer L7cy**
>
> References:
>
> 	[1]. Garg, Saurabh; Wu, Yifan; Balakrishnan, Sivaraman; Lipton, Zachary C. (2020). “A unified view of label shift estimation”.
> 	[2]. Alexandari, Amr; Kundaje, Anshul; Shrikumar, Avanti (2020). “Maximum Likelihood with Bias-Corrected Calibration is Hard-To-Beat at Label Shift Adaptation”.
> 	[3]. Guo, Chuan; Pleiss, Geoff; Sun, Yu; Weinberger, Kilian Q. (2017). “On Calibration of Modern Neural Networks”.
> 	[4]. Wang, Cheng (2024). “Calibration in Deep Learning: A Survey of the State-of-the-Art”.
> 	[5]. Pereyra, Gabriel, Tucker, George, Chorowski, Jan, Kaiser, Łukasz, and Hinton, Geoffrey (2017). “Regularizing Neural Networks by Penalizing Confident Output Distributions”.
> 	[6]. Müller, Rafael, Kornblith, Simon, and Hinton, Geoffrey (2020). “When Does Label Smoothing Help?”.
> 	[7]. Kumar, Aviral, Sarawagi, Sunita, and Jain, Ujjwal (2018). “Trainable Calibration Measures for Neural Networks from Kernel Mean Embeddings.”
> 	[8]. Mukhoti, Jishnu, Kulharia, Viveka, Sanyal, Amartya, Golodetz, Stuart, Torr, Philip H. S., and Dokania, Puneet K. (2020). “Calibrating Deep Neural Networks using Focal Loss.”
>     [9]. McMahan, H. B., Moore, E., Ramage, D., Hampson, S., & Agüera y Arcas, B. (2023). Communication-Efficient Learning of Deep Networks from Decentralized Data.
>     [10]. K. Luo, X. Li, Y. Lan and M. Gao, "GradMA: A Gradient-Memory-based Accelerated Federated Learning with Alleviated Catastrophic Forgetting."
>     [11]. Kim, J., Kim, G., & Han, B. (2022). Multi-Level Branched Regularization for Federated Learning.
>     [12]. McMahan, Brendan, et al. "Communication-efficient learning of deep networks from decentralized data." Artificial intelligence and statistics. PMLR, 2017.
>     [13]. Neo, D., Winkler, S., & Chen, T. (2024). MaxEnt Loss: Constrained Maximum Entropy for Calibration under Out-of-Distribution Shift.

---

> ### Author Response · Authors · 2024-11-23
> **Follow-up Discussion**
>
> Dear Reviewer L7cy, we hope we have addressed all of your questions and concerns. In summary, we clarified the concept of “underlying uncertainty” and its connection to entropy regularization within the context of our IW-ERM framework with VRLS. Additionally, we conducted experiments in distributed learning, comparing our method to the latest baselines under various carefully designed scenarios, demonstrating the superiority of our approach. We look forward to continuing this discussion and are happy to provide any further explanations or details if needed.

---

> > ### Comment · Reviewer_L7cy · 2024-11-25
> >
> > Thank you for your response. The new explanations and experimental results have strengthened the evaluation, and I believe the manuscript refinements will enhance the paper's ability to effectively convey key information. I would like to revise my rating, acknowledging the authors' efforts, and weakly supporting the acceptance of this paper.

---

> > > ### Author Response · Authors · 2024-11-25
> > > **Response to Reviewer L7cy**
> > >
> > > We sincerely appreciate the time and effort you have dedicated to re-evaluating our work and providing valuable feedback that has contributed to its improvement.

---

### Official Review · Reviewer_Ym52 · 2024-10-29

**Soundness:** 3
**Presentation:** 2
**Contribution:** 3
**Rating:** 6
**Confidence:** 4

**Summary:**

This paper addresses the challenge of inter-node and intra-node label shifts in distributed learning. Test-to-train density ratio estimation plays an important role in addressing the label-shift problem. The authors proposed the Versatile Robust Label Shift (VRLS) method, which incorporates a regularization term based on Shannon entropy. This regularization term significantly improves the estimation of the conditional distribution $p^{tr}(\mathbf{y}|\mathbf{x})$, thereby enhancing the estimation of the label density ratio. The authors conduct experiments on MNIST, Fashion-MNIST, and CIFAR-10 datasets to verify the effectiveness of the proposed method. The authors also provide theoretical analysis for the estimation error and convergence.

**Strengths:**

1. The authors proposed a simple but effective method to address the problem of label shift in distributed learning.
2. The authors conduct experiments to verify their proposed method. The MSE for the density ratio of the proposed method is lower than that of baselines. The test accuracy of the proposed method is also better than that of baselines.
3. Bounds on ratio estimation and convergence rates are analyzed theoretically.

**Weaknesses:**

1. The definition of inter-node and intra-node label shifts is ambiguous. I suggest that the authors define them more clearly.
2. In Lines 196-198, why penalizing the Shannon entropy of the softmax output can improve the approximation of $p^{tr}(\mathbf{y}|\mathbf{x})$? In other words, why the smoother output is a better approximation of $p^{tr}(\mathbf{y}|\mathbf{x})$ (Line 207-208)?
3. What is the meaning of $\lambda_k$ in the risk of IW-ERM?
4. Lack of sensitivity test for the hyperparameter $\zeta$ in Equation (2).
5. The font sizes in Fig 1 and Fig 2 are inconsistent.

**Questions:**

See weaknesses.

---

> ### Author Response · Authors · 2024-11-21
> **Response to Reviewer Ym52**
>
> **1. The definition of inter-node and intra-node label shifts is ambiguous. I suggest that the authors define them more clearly.**
>
>
> In our distributed learning scenario with $K$ nodes, we define intra-node label shift as the scenario where, for any given node $k$, the label distribution differs between its training and test sets, i.e., $p_k^{tr}(y) \neq p_k^{te}(y)$. Conversely, inter-node label shift refers to the situation where the label distributions differ across nodes, such that for any two nodes $k$ and $j$, $p_k^{tr}(y) \neq p_j^{te}(y)$. For further clarification, Table 4 and Appendix C provide detailed illustrations of these definitions.
>
> 2. **In Lines 196-198, why penalizing the Shannon entropy of the softmax output can improve the approximation of ? In other words, why the smoother output is a better approximation of $p_{tr}(\textbf{y}|\textbf{x})$ (Line 207-208)?**
>
>
> Softmax aggregates the logits and provides a probability distribution over the classes. However, it does not inherently address overconfidence, as it simply scales the logits without ensuring they reflect the true underlying distribution. Entropy regularization, as outlined in Common Response 1, plays a critical role in penalizing **low-entropy** predictions, encouraging the model to learn to represent uncertainty more faithfully, and reducing the risk of overconfident or underconfident outputs [3, 4]. It is well known that entropy regularization mitigates overfitting to the negative log-likelihood (NLL), which evaluates the quality of probabilistic predictions and often correlates with low-entropy outputs [2–9]. Since we approximate $p_{tr}(\textbf{y}|\textbf{x})$, promoting smoother predictions can avoid extreme probabilities and aligns more closely with the true distribution, improving its calibration and robustness under label shift.
>
> 3. **What is the meaning of $\lambda_k$ in the risk of IW-ERM?**
>
> In the original IW-ERM, the local coefficients $\lambda_k$'s  satisfy the conditions $\lambda_k \geq 0$ for all $k$ and $\sum_{k=1}^{K} \lambda_k = 1$. For any such $\lambda_k$'s, the IW-ERM provides a consistent estimate of the minimizer of the overall true risk $\sum_{k=1}^K R_k$, as shown in Proposition 4.1. To simplify the formulation and improve the readability of our work, **in our revision**, we simply use uniform weights, $\lambda_k = 1/K$, in the formulation and experiments as we do not need any additional tunable hyperparameters and our results hold without any additinal tuning.
>
> 4. **Lack of sensitivity test for the hyperparameter $\zeta$ in Equation (2).**
>
> We acknowledge that a detailed sensitivity analysis for the hyperparameter $\zeta$ in Equation (2) is missing from our current submission. However, we conducted preliminary tests and found that the recommended values from [5]—specifically $\zeta = 1.0$ for MNIST and $\zeta = 0.1$ for CIFAR-10—consistently produced stable and robust results across our experiments.
>
>
>
> 5.**The font sizes in Fig 1 and Fig 2 are inconsistent.**
>
> We ensured consistent font sizes within each figure by optimizing for readability based on the available space and detail in each subfigure. After reordering and resizing, all subfigures within each figure now have the same font size. Meanwhile, this adjustment re-organize the experimental results, Figure 1 for MSE vs. label shift parameters across datasets and relaxed label shift conditions, and Figure 2 MSE vs. test sample sizes.
>
> References:
>
> 	[1]. Garg, Saurabh; Wu, Yifan; Balakrishnan, Sivaraman; Lipton, Zachary C. (2020). “A unified view of label shift estimation”.
> 	[2]. Alexandari, Amr; Kundaje, Anshul; Shrikumar, Avanti (2020). “Maximum Likelihood with Bias-Corrected Calibration is Hard-To-Beat at Label Shift Adaptation”.
> 	[3]. Guo, Chuan; Pleiss, Geoff; Sun, Yu; Weinberger, Kilian Q. (2017). “On Calibration of Modern Neural Networks”.
> 	[4]. Wang, Cheng (2024). “Calibration in Deep Learning: A Survey of the State-of-the-Art”.
> 	[5]. Pereyra, Gabriel, Tucker, George, Chorowski, Jan, Kaiser, Łukasz, and Hinton, Geoffrey (2017). “Regularizing Neural Networks by Penalizing Confident Output Distributions”.
> 	[6]. Müller, Rafael, Kornblith, Simon, and Hinton, Geoffrey (2020). “When Does Label Smoothing Help?”.
> 	[7]. Kumar, Aviral, Sarawagi, Sunita, and Jain, Ujjwal (2018). “Trainable Calibration Measures for Neural Networks from Kernel Mean Embeddings.”
> 	[8]. Mukhoti, Jishnu, Kulharia, Viveka, Sanyal, Amartya, Golodetz, Stuart, Torr, Philip H. S., and Dokania, Puneet K. (2020). “Calibrating Deep Neural Networks using Focal Loss.”
>     [9]. Neo, D., Winkler, S., & Chen, T. (2024). MaxEnt Loss: Constrained Maximum Entropy for Calibration under Out-of-Distribution Shift.

---

> ### Author Response · Authors · 2024-11-23
> **Follow-up Discussion**
>
> Dear Reviewer Ym52, we hope we have addressed all of your questions and concerns. In brief, we clarified the definitions of intra- and inter-node label shifts, provided an explanation of Shannon entropy regularization, smooth predictions, and their role in improving the approximation of the true conditional distribution. Additionally, we clarified the meaning of the hyperparameters in IW-ERM and adjusted the figures accordingly. We look forward to continuing this discussion and are happy to provide any further explanations or details if needed.

---

> > ### Comment · Reviewer_Ym52 · 2024-11-27
> >
> > Thank you for the response and revision. Based on your new explanation, I understand that entropy regularization effectively addresses the issue of over-confidence resulting from training with cross-entropy loss. By calibrating the neural network outputs, entropy regularization enhances the estimation of the conditional distribution $p^{tr}(y|x)$, leading to a more accurate estimation of the ratio $r(y)$. Given that all my initial concerns have been addressed, I have decided to revise my rating.

---

> > > ### Author Response · Authors · 2024-11-27
> > > **Response to Reviewer Ym52**
> > >
> > > We are truly grateful for the time and effort you invested in reviewing our work again, as well as for your thoughtful feedback, which has greatly improved its quality.

---

### Official Review · Reviewer_4bHP · 2024-11-03

**Soundness:** 3
**Presentation:** 3
**Contribution:** 3
**Rating:** 8
**Confidence:** 3

**Summary:**

The paper proposes the Versatile Robust Label Shift (VRLS) method to address label shift in distributed learning systems. VRLS employs entropy regularization to improve density ratio estimation between training and test label distributions, making it resilient to both intra-node and inter-node label shifts. The paper further extends VRLS to a multi-node setup through an Importance Weighted ERM (IW-ERM) framework, which balances label distribution estimation across nodes while preserving data privacy. Empirical results show VRLS outperforming baselines by up to 20% in imbalanced settings, and theoretical bounds confirm VRLS's estimation accuracy and convergence rates.

**Strengths:**

1-VRLS introduces a unique entropy regularization term that enhances density ratio estimation by penalizing degenerate solutions, thus yielding well-calibrated predictions. Unlike existing methods such as MLLS and BBSE, which do not adequately capture uncertainty, VRLS adapts the estimation process to real-world variability. This improvement aligns with recent work in calibration under distribution shifts (e.g., Guo et al., 2017), offering both theoretical and practical robustness.

2-The paper’s thorough empirical evaluation spans single-node and multi-node scenarios, with tests on MNIST, Fashion MNIST, and CIFAR-10. In distributed settings, VRLS with IW-ERM adapts effectively, providing significant accuracy improvements across nodes while handling inter-node and intra-node label shifts. Compared to methods such as FedAvg and FedProx, which struggle with label shift, VRLS maintains robust performance in both settings, a key distinction from prior approaches.

3- Theoretical error bounds and convergence rates for VRLS are rigorously established. The authors show that the VRLS method’s estimation error decreases proportionally to the square root of test sample size, a valuable addition to the literature where empirical results often lack formal guarantees. This theoretically grounded approach helps ensure VRLS's performance consistency in both high and low label-shift scenarios, a substantial contribution over prior studies that assume stability without formal validation.

4- By using locally computed density ratios without requiring data exchange, IW-ERM maintains data privacy across nodes, a marked improvement over traditional federated approaches. This reduction in communication rounds, coupled with enhanced convergence rates, renders IW-ERM with VRLS practical for real-world applications where privacy and resource efficiency are paramount, setting it apart from resource-intensive federated models like FedBN.

**Weaknesses:**

1- The paper effectively shows VRLS's reliance on Shannon entropy regularization but does not explore how it compares to other regularization techniques, such as focal loss or maxent loss. A comparative analysis across regularization methods on the same datasets, particularly with focal loss regularization, would help clarify VRLS's robustness and calibration capabilities. Additionally, a theoretical discussion on the effects of different regularizations on model calibration and uncertainty estimation could deepen the understanding of VRLS's effectiveness.

2-VRLS primarily addresses scenarios under symmetric or Dirichlet-based label shifts, with limited discussion on its performance with highly non-uniform or asymmetric label shifts. Extending empirical analysis to include more diverse label distributions would provide a clearer picture of VRLS’s generalizability to complex, real-world shifts.

3- The theoretical error bounds rely on assumptions that may not fully align with real-world distributed data characteristics, such as bounded data spaces and sub-Gaussian noise. Providing examples of scenarios where these assumptions might fail, along with discussing VRLS’s potential performance in such cases, would add rigor to the theoretical claims. Additionally, testing VRLS with artificially unbounded or heavy-tailed data distributions would offer practical insights into the method’s robustness beyond the idealized assumptions, adding depth to its theoretical grounding.

4- The paper’s focus on label shift in classification limits its applicability to other supervised learning tasks, such as regression or multi-label classification, where label distributions differ. Addressing how VRLS might generalize to other tasks, or clarifying any modifications needed, could extend the method’s relevance across broader distributed learning applications.

5- The VRLS method’s reliance on entropy-based regularization (inspired by Shannon entropy) is theoretically motivated by preventing degenerate solutions. However, there is limited theoretical analysis on whether this regularization could inadvertently introduce biases, especially in settings where label shifts deviate from the assumptions about class-conditional stability. This could be explored further in the theoretical results or discussed as a potential limitation.

**Questions:**

1-  Could the authors discuss how VRLS would perform with other regularization methods, such as focal loss, or compare its performance with maxent loss for calibration? Would such alternatives affect calibration or robustness?

2- Can the authors provide empirical or theoretical evidence on VRLS’s performance under non-uniform or more complex label shift distributions, such as class-conditional label shift?

3- How does VRLS handle scalability in networks with significantly more nodes or nodes with varying computational capacities? Are there adjustments to VRLS or IW-ERM to address such scenarios?

4-  Given that assumptions like bounded data and sub-Gaussian noise are not always valid in practice, could the authors discuss how VRLS might be adapted or how results may vary when these assumptions do not hold?

5-  VRLS is designed for classification, but can it be adapted for regression tasks where label distributions may also shift? If so, what adjustments would be needed?

---

> ### Author Response · Authors · 2024-11-21
> **Response to Reviewer 4bHP**
>
> 1. **Could the authors discuss how VRLS would perform with other regularization methods, such as focal loss, or compare its performance with maxent loss for calibration? Would such alternatives affect calibration or robustness?**
>
>
> Although our primary focus is on addressing label shift within a multi-node learning environment via the IW-ERM framework, exploring the performance of VRLS with other regularizers, particularly within the IW-ERM context, is an intriguing research direction. While we could not ablate focal loss [1] or maxent loss [2] within our current framework due to time constraints, these methods share conceptual similarities with ours within the principle of maximum entropy (Further discussion in Common Response 1). Focal loss can be viewed as a variant of this principle, where it balances KL divergence and entropy, aiming to improve calibration. Maxent loss [2] extends this concept by incorporating additional constraints on the mean and variance of predictions, making it particularly effective in scenarios involving out-of-distribution (OOD) data. The regularizer used in VRLS is conceptually close to the variance constraints employed in MaxEnt loss, but it differs in the specific deviation metrics utilized.
>
> 2. **Can the authors provide empirical or theoretical evidence on VRLS’s performance under non-uniform or more complex label shift distributions, such as class-conditional label shift?**
>
> We have conducted experiments under generalized or relaxed label shift scenarios, including non-stationary class-conditional distributions $p(x|y)$, where changes in both $p(y)$ and $p(x|y)$ contribute to overall distribution shifts. The results, presented in Figure 1 and Figure 2, demonstrate the superior performance of VRLS compared to MLLS [3] in these complex scenarios.
>
> Although VRLS is not explicitly designed to address class-conditional label shift, its combination of regularized training with cross-entropy loss and an entropy regularizer equips it with inherent robustness against such distributional changes. The entropy regularizer, by promoting calibrated predictions and enhancing out-of-distribution (OOD) generalization [2, 4, 5], allows VRLS to adapt effectively to these dynamic and challenging shifts.

---

> ### Author Response · Authors · 2024-11-21
> **Response to Reviewer 4bHP**
>
> 3. **How does VRLS handle scalability in networks with significantly more nodes or nodes with varying computational capacities? Are there adjustments to VRLS or IW-ERM to address such scenarios?**
>
> As the number of nodes increases, the primary challenge lies in efficiently estimating density ratios on each node. VRLS is robust in this regard, as demonstrated in Figure 2, which shows low estimation error even with fewer test samples. This allows for reduced test sample usage without significantly compromising accuracy. Additionally, in the aggregation phase (Phase 2 of Algorithm 2), the aggregated estimates on each node makes it more robust to individual inaccuracies, further enhancing scalability.
>
> Another concern is the computational effort required to train individual predictors on each node. To address this, a single shared predictor can be trained centrally and then distributed to all nodes, reducing the need for separate predictors on each node and lowering the overall computational load. These adjustments help ensure that VRLS stays efficient and scalable, even in networks with more nodes or varying computational capabilities.
>
> 4. **Given that assumptions like bounded data and sub-Gaussian noise are not always valid in practice, could the authors discuss how VRLS might be adapted or how results may vary when these assumptions do not hold?**
>
>
> In practice, sub-Gaussian noise assumption is satisfied under common deep learning practice such as gradient clipping and normalization techniques, which effectively handles the presence of heavy-tailed or non-sub-Gaussian noise by limiting the impact of extreme gradients. Please kindly note that we considered sub-Gaussian noise only for High-probability Bound for Nonconvex Optimization in Theorem 5.5. Our other convergence results do not require such assumption.
>
> We agree developing bounds under relaxed assumptions such as Hölder continuity instead of global boundedness will be very interesting as future work.
>
> 5. **VRLS is designed for classification, but can it be adapted for regression tasks where label distributions may also shift? If so, what adjustments would be needed?**
>
> VRLS, build on Maximum Likelihood Estimation (MLE) and combined with entropy maximization, is primarily designed for classification. While its framework for estimating conditional distributions $p_{tr}(y|x)$ could extend to regression tasks, its current formulation does not directly support this. In regression, uncertainty is represented through prediction intervals rather than confidence scores, limiting the relevance of maximum entropy methods. However, VRLS’s focus on accurate conditional probability approximation remains applicable and could integrate techniques like quantile regression [6, 7] or Bayesian methods, though such extensions fall outside the current scope.
>
>
> References:
>
> 	 [1]. Mukhoti, Jishnu, Kulharia, Viveka, Sanyal, Amartya, Golodetz, Stuart, Torr, Philip H. S., and Dokania, Puneet K. (2020). “Calibrating Deep Neural Networks using Focal Loss.”
>      [2]. Neo, D., Winkler, S., & Chen, T. (2024). MaxEnt Loss: Constrained Maximum Entropy for Calibration under Out-of-Distribution Shift.
>      [3]. Garg, Saurabh; Wu, Yifan; Balakrishnan, Sivaraman; Lipton, Zachary C. (2020). “A unified view of label shift estimation”.
>      [4]. Zhao, Shanshan, Gong, Mingming, Liu, Tongliang, Fu, Huan, and Tao, Dacheng. (2020). “Domain Generalization via Entropy Regularization.”
>      [5]. Schoots, Nandi, and Dylan Cope. (2023). “Low-Entropy Latent Variables Hurt Out-of-Distribution Performance.”
>      [6]. Hardin, James, Henrik Schmeidiche, and Raymond Carroll. (2003). “The Regression-calibration Method for Fitting Generalized Linear Models with Additive Measurement Error.”
>      [7]. Song, Hao, Tom Diethe, Meelis Kull, and Peter Flach. (2019). “Distribution Calibration for Regression.”

---

> ### Author Response · Authors · 2024-11-23
> **Follow-up Discussion**
>
> Dear Reviewer 4bHP, we hope we have addressed all of your questions and concerns. In summary, we conducted experiments on ImageNet-Tiny and provided a detailed explanation of how our entropy regularization compares with other regularization methods. Additionally, we discussed the scalability of IW-ERM with VRLS, VRLS’s potential extension to regression tasks, and its applicability beyond the sub-Gaussian noise assumption. We look forward to continuing this discussion and are happy to provide any additional explanations or details if needed.

---

> > ### Comment · Reviewer_4bHP · 2024-12-02
> >
> > The reviewer thanks the authors for the response -- they addressed the concerns and the reviewer maintains the positive rating.

---

> > > ### Author Response · Authors · 2024-12-02
> > > **Response to Reviewer 4bHP**
> > >
> > > We sincerely value the thoughtful attention and effort you have given to our submission and for providing positive and insightful feedback, along with your recognition of its value, which has been both encouraging and motivating for us.

---

### Official Review · Reviewer_og3p · 2024-11-04

**Soundness:** 3
**Presentation:** 2
**Contribution:** 2
**Rating:** 6
**Confidence:** 4

**Summary:**

This paper solves the problem where distributed data nodes have different label distributions in their training and test data. The authors propose VRLS - a method that estimates how these distributions differ between nodes and use this information to improve model training. Their approach maintains data privacy while achieving 20% better accuracy than existing methods.

**Strengths:**

1. This paper clearly articulates its research focus by addressing both inter-node and intra-node label shifts in distributed learning systems, which represents a more comprehensive approach than prior works.
2. The paper provides extensive theoretical proofs to support the proposed algorithm, though I have not thoroughly examined all of them.

**Weaknesses:**

1. The paper lacks clear logical flow and detailed algorithm explanations. While the title emphasizes distributed learning, Section 4 (covering VLRS in distributed environments) is surprisingly brief - only one-third of a page. It simply states that minimizing equation (IW_ERM) enables VLRS in distributed settings, without proper elaboration. In contrast, Section 5 on VLRS bounds spans two full pages. It's puzzling why the authors prioritized theoretical bounds over making their work more accessible to readers.
2. Lines 193-194 contain an inconsistency where the authors claim "Our proposed VRLS objective aims to better approximate the conditional distribution ptr(y|x)", but equation 1 (VRLS objective) actually estimates r. The conditional distribution approximation corresponds to equation 2.
3. The authors fail to adequately explain why minimizing equation 2 leads to better conditional distribution estimation. While Ω(fθ) represents entropy and promotes more dispersed neural network outputs as claimed, there's no clear justification for how this ensures f(x) better estimates ptr(y|x).
4. The paper suffers from poor notation management. Symbols like Ω(fθ) should be defined at first appearance rather than later, as this significantly increases the time needed to understand the paper.
5. The experimental validation is unconvincing, relying only on synthetic datasets (MNIST and CIFAR-10) with small images and just 10 classes. The evaluation should include larger datasets like ImageNet and real-world scenarios. The lack of real-world validation raises concerns about whether the paper's assumptions are practical.

**Questions:**

same as the third point of weakness: why minimizing equation 2 leads to better conditional distribution estimation?

---

> ### Author Response · Authors · 2024-11-21
> **Response to Reviewer og3p**
>
> 1. **The paper lacks clear logical flow and detailed algorithm explanations. While the title emphasizes distributed learning, Section 4 (covering VLRS in distributed environments) is surprisingly brief - only one-third of a page. It simply states that minimizing equation (IW_ERM) enables VLRS in distributed settings, without proper elaboration. In contrast, Section 5 on VLRS bounds spans two full pages. It's puzzling why the authors prioritized theoretical bounds over making their work more accessible to readers.**
>
> Thank you for your constructive feedback. To address your comment on Section 4, we have significantly expanded this section to provide a more detailed explanation of VLRS in distributed environments. The revised content (lines 208–258, highlighted in blue) elaborates on the extension of VLRS to multi-node settings under the IW-ERM framework. Additionally, Algorithm 2 (L216–231) now includes the algorithmic descriptions, while Table 4 (L108–115) details scenarios involving intra- and inter-label shifts.
>
> Section 5 serves two critical purposes: Theorem 5.1 establishes the density ratio estimation error bound for a single node, which forms the statistical foundation of VRLS. Theorems 5.2–5.5 extend this to distributed learning, providing communication guarantees and convergence analysis with IW-ERM. These bounds are essential to validating our method’s robustness and effectiveness. Section 5, together with the revised Section 4, clearly demonstrates the connection between the algorithmic framework and its theoretical guarantees.
>
> 2. **Lines 193-194 contain an inconsistency where the authors claim "Our proposed VRLS objective aims to better approximate the conditional distribution $p_{tr}(y|x)$", but equation 1 (VRLS objective) actually estimates $r$. The conditional distribution approximation corresponds to equation 2.**
>
> The primary objective of VRLS is to estimate the density ratio $r$, as defined in Equation (3) (L179 of the revised manuscript). This estimation is achieved using a predictor $f_\theta(x)$, which is trained within the regularized framework described in Equation (4) (L182 of the revised manuscript) using the training data. The predictor $f_\theta(x)$ approximates the conditional distribution [1], which is essential for estimating $r$. This process is thoroughly explained in the revised manuscript in L147–150.
>
> 3.**The authors fail to adequately explain why minimizing equation 2 leads to better conditional distribution estimation. While $\Omega(f_θ)$ represents entropy and promotes more dispersed neural network outputs as claimed, there's no clear justification for how this ensures $f(x)$ better estimates $p_{tr}(y|x)$.**
>
>
> Thank you for this insightful comment. As summarized in Common Response 1 and detailed in the revised manuscript (L151–155), addressing label shift requires accurate estimation of the conditional distribution $p_{tr}(y|x)$ with neural network $f_{\theta}(x)$. This task is inherently difficult due to finite-sample and miscalibration errors, as described in Theorem 3 of [1]. Overfitting to NLL, as highlighted in [2–8], is associated with low entropy, leading to miscalibrated predictions. Our work **explicitly** incorporates an entropy regularizer into the training objective to mitigate such overfitting, addressing miscalibration and aligning predicted probabilities with true likelihoods.
>
> Minimizing Equation (2) integrates cross-entropy loss and entropy regularization, leveraging principles of entropy maximization from methods like focal loss and post-hoc calibration techniques [2–8]. By promoting higher entropy, this approach ensures predictions are better calibrated and more accurately approximate $p_{tr}(y|x)$. Validated both theoretically and empirically in [1] and related works [2-8], VRLS combines entropy-based calibration into training, achieving superior estimation performance under diverse label shift scenarios while improving overall alignment with the true conditional distribution.
>
> 4. **The paper suffers from poor notation management. Symbols like Ω(fθ) should be defined at first appearance rather than later, as this significantly increases the time needed to understand the paper.**
>
>
> Thank you for highlighting this issue with notation. In the revision, we have adjusted the presentation order in the revised manuscript so that $\Omega(f_\theta)$ is defined at their first appearance in Equation (2) in L158-160, making sure that this notation has been introduced clearly before being used in the context of training the classifier.

---

> ### Author Response · Authors · 2024-11-21
> **Response to Reviewer og3p**
>
> 5. **The experimental validation is unconvincing, relying only on synthetic datasets (MNIST and CIFAR-10) with small images and just 10 classes. The evaluation should include larger datasets like ImageNet and real-world scenarios. The lack of real-world validation raises concerns about whether the paper's assumptions are practical.**
>
> We understand your concern. Creating realistic label shift scenarios is indeed challenging when using publicly available datasets, as most of them have balanced class distributions by design. In this work, we followed the conventional but intentional disruptions to simulate label shift. We systematically vary label shift severities and sample sizes, by sampling 100 subsets per severity level, across additional datasets like ImageNet-Tiny and Oxford 102 during the rebuttal phase. On ImageNet-Tiny, VRLS outperformed MLLS by 2.8% (see table), demonstrating its robustness in complex label shift scenarios. On Oxford 102, which includes 102 classes and an imbalanced test set, VRLS achieved comparable estimation errors to MLLS (~2.5×10−3), further validating its effectiveness. Notably, unlike the other datasets, Oxford 102 dataset holds natural label shift, and we used the inherent label shift rather than simulating varying scenarios. While VRLS demonstrated comparable performance to MLLS on this dataset, the fixed severity and label distribution inherent to Oxford 102 limit its ability to generalize to broader scenarios. Further response to this comment is addressed in Common Response 2.
>
> In addtion, we compared with our IW-ERM with VRLS with additional baselines GradMA [10], FedMLB [11] on binary classification datasets carefully designed to reflect varying levels of classification difficulty. These datasets feature controlled adjustments in noise levels, covariance structures, and sample overlap, simulating diverse and realistic classification scenarios. Results show IW-ERM consistently outperforms FedAvg, FedMLB and GradMA across all datasets, demonstrating its statistical robustness and effectiveness under label shift. Further response to this comment is addressed in Common Response 3.
>
> | Dataset | Method | Avg. MSE       |
> |---------|--------|----------------|
> | ImageNet-Tiny    | MLLS | 5.8068e-03 |
> | ImageNet-Tiny    | VRLS | **5.6467e-03** |
>
>
> | **Datasets** | **Method**       | **Accuracy (%)** |
> |--------------|------------------|------------------|
> | **Dataset 1** | FedAvg           | 59.38           |
> |               | FedMLB           | 62.50           |
> |               |    GradMA     |       63.62     |
> |               | IW-ERM (Ours)    | **65.88**       |
> | **Dataset 2** | FedAvg           | 60.50           |
> |               | FedMLB           | 62.25           |
> |               |    GradMA     |      62.38     |
> |               | IW-ERM (Ours)    | **63.50**       |
> | **Dataset 3** | FedAvg           | 62.50           |
> |               | FedMLB           | 64.00           |
> |               |   GradMA      |      64.62      |
> |               | IW-ERM (Ours)    | **65.63**       |
>
> References:
>
> 	[1]. Garg, Saurabh; Wu, Yifan; Balakrishnan, Sivaraman; Lipton, Zachary C. (2020). “A unified view of label shift estimation”.
> 	[2]. Alexandari, Amr; Kundaje, Anshul; Shrikumar, Avanti (2020). “Maximum Likelihood with Bias-Corrected Calibration is Hard-To-Beat at Label Shift Adaptation”.
> 	[3]. Guo, Chuan; Pleiss, Geoff; Sun, Yu; Weinberger, Kilian Q. (2017). “On Calibration of Modern Neural Networks”.
> 	[4]. Wang, Cheng (2024). “Calibration in Deep Learning: A Survey of the State-of-the-Art”.
> 	[5]. Pereyra, Gabriel, Tucker, George, Chorowski, Jan, Kaiser, Łukasz, and Hinton, Geoffrey (2017). “Regularizing Neural Networks by Penalizing Confident Output Distributions”.
> 	[6]. Müller, Rafael, Kornblith, Simon, and Hinton, Geoffrey (2020). “When Does Label Smoothing Help?”.
> 	[7]. Kumar, Aviral, Sarawagi, Sunita, and Jain, Ujjwal (2018). “Trainable Calibration Measures for Neural Networks from Kernel Mean Embeddings.”
> 	[8]. Mukhoti, Jishnu, Kulharia, Viveka, Sanyal, Amartya, Golodetz, Stuart, Torr, Philip H. S., and Dokania, Puneet K. (2020). “Calibrating Deep Neural Networks using Focal Loss.”
>     [9]. McMahan, H. B., Moore, E., Ramage, D., Hampson, S., & Agüera y Arcas, B. (2023). Communication-Efficient Learning of Deep Networks from Decentralized Data.
>     [10]. K. Luo, X. Li, Y. Lan and M. Gao, "GradMA: A Gradient-Memory-based Accelerated Federated Learning with Alleviated Catastrophic Forgetting."
>     [11]. Kim, J., Kim, G., & Han, B. (2022). Multi-Level Branched Regularization for Federated Learning.

---

> ### Author Response · Authors · 2024-11-23
> **Follow-up Discussion**
>
> Dear Reviewer og3p, we hope we have addressed all of your questions and concerns. In brief, we have significantly expanded Section 4 to provide a more detailed explanation of VLRS in distributed environments. Additionally, we conducted experiments on ImageNet-Tiny and provided a comprehensive explanation of the relationship between entropy regularization and the true conditional distribution. We look forward to furthering this discussion and are happy to provide any additional explanations or details as needed.

---

> ### Author Response · Authors · 2024-11-28
> **Further Key Clarifications**
>
> Dear Reviewer og3p,
>
> Thank you once again for your valuable feedback! Following your suggestions, we have conducted additional experiments using a real-world multi-class dataset (ImageNet-Tiny) to enhance the practical relevance of our work. More importantly, we have restructured Section 4 to provide a clearer explanation of IW-ERM, VRLS, and the relationship with Shannon entropy. Please let us know if there is anything further you would like us to address. We appreciate your time and consideration.

---

### Author Response · Authors · 2024-11-21
**Common Response**

We received thoughtful comments. Addressing them has improved the paper considerably. We now summarize the key and common comments we have addressed.

**1.Explanation of Entropy Regularization:**

Entropy regularization encourages the model's output probability distribution to align more closely with its inherent true distribution, helping to reduce overconfidence and improve calibration. Prior work [2–8] improves calibration through post-hoc [3] and regularization approaches [5, 6, 8, 15], based on entropy maximization principles. For example, focal loss $\mathcal{L}_f$ [6] balances KL divergence $\text{KL}(q \parallel \hat{p})$ (where $q$ is the true label distribution and $\hat{p}$ is the predicted distribution) and prediction entropy $\mathbb{H}[\hat{p}]$ using a hyperparameter $\gamma$:

$$
\mathcal{L}_f \geq \text{KL}(q \parallel \hat{p}) + \mathbb{H}[q] - \gamma \mathbb{H}[\hat{p}].
$$

Our VRLS is the first to apply an **explicit** entropy regularizer for the label shift problem. Building on MLLS [1], which attributes errors to finite-sample and miscalibration effects, VRLS applies maximum entropy principles to address calibration errors and improve ratio estimation for label shift. Backed by theoretical guarantees (Theorems 5.1-5.5) and extensive experiments (Figures 1-2), VRLS consistently outperforms baselines across diverse label shift severities and test set sizes.


**2.Experimental Validation in Realistic Settings:**

Creating realistic settings is challenging due to the inherent class balance in most public datasets, requiring intentional disruptions to simulate label shift. Our experiments address this by systematically varying label shift severities and sample sizes, by sampling 100 subsets per severity level, across datasets like CIFAR-10, MNIST, and FashionMNIST, with further validation using ImageNet-Tiny and Oxford 102 during the rebuttal phase. On ImageNet-Tiny, VRLS outperformed MLLS by 2.8% (see table), demonstrating its robustness in complex label shift scenarios. On Oxford 102, which includes 102 classes and an imbalanced test set, VRLS achieved comparable estimation errors to MLLS (~2.5×10−3), further validating its effectiveness. Notably, unlike the other datasets, Oxford 102 dataset holds natural label shift, and we used the inherent label shift rather than simulating varying scenarios. While VRLS demonstrated comparable performance to MLLS on this dataset, the fixed severity and label distribution inherent to Oxford 102 limit its ability to generalize to broader scenarios.


| Dataset | Method | Avg. MSE       |
|---------|--------|----------------|
| ImageNet-Tiny    | MLLS | 5.8068e-03 |
| ImageNet-Tiny    | VRLS | **5.6467e-03** |

**3.Comparison with (Luo et al.[10]; Kim et al.[11]):**


The suggested baselines employ ERM for distributed learning, addressing client heterogeneity through hybrid subnetworks [11] and gradient memory with quadratic constraints [10]. However, they do not account for label shift with statistical guarantees. Our IW-ERM approach, specifically designed for label shift, fills this gap. We compared our IW-ERM with FedAvg [12], GradMA [10], and FedMLB [11] on binary classification datasets carefully designed to reflect varying levels of classification difficulty. These datasets feature controlled adjustments in noise levels, covariance structures, and sample overlap, simulating diverse and realistic classification scenarios. Results show IW-ERM consistently outperforms FedAvg, FedMLB and GradMA across all datasets, demonstrating its statistical robustness and effectiveness under label shift.

| **Datasets** | **Method**       | **Accuracy (%)** |
|--------------|------------------|------------------|
| **Dataset 1** | FedAvg           | 59.38           |
|               | FedMLB           | 62.50           |
|               |    GradMA     |       63.62     |
|               | IW-ERM (Ours)    | **65.88**       |
| **Dataset 2** | FedAvg           | 60.50           |
|               | FedMLB           | 62.25           |
|               |    GradMA     |      62.38     |
|               | IW-ERM (Ours)    | **63.50**       |
| **Dataset 3** | FedAvg           | 62.50           |
|               | FedMLB           | 64.00           |
|               |   GradMA      |      64.62      |
|               | IW-ERM (Ours)    | **65.63**       |

---

> ### Author Response · Authors · 2024-11-21
> **References of Common Response**
>
> References:
>
> 	[1]. Garg, Saurabh; Wu, Yifan; Balakrishnan, Sivaraman; Lipton, Zachary C. (2020). “A unified view of label shift estimation”.
> 	[2]. Alexandari, Amr; Kundaje, Anshul; Shrikumar, Avanti (2020). “Maximum Likelihood with Bias-Corrected Calibration is Hard-To-Beat at Label Shift Adaptation”.
> 	[3]. Guo, Chuan; Pleiss, Geoff; Sun, Yu; Weinberger, Kilian Q. (2017). “On Calibration of Modern Neural Networks”.
> 	[4]. Wang, Cheng (2024). “Calibration in Deep Learning: A Survey of the State-of-the-Art”.
> 	[5]. Pereyra, Gabriel, Tucker, George, Chorowski, Jan, Kaiser, Łukasz, and Hinton, Geoffrey (2017). “Regularizing Neural Networks by Penalizing Confident Output Distributions”.
> 	[6]. Müller, Rafael, Kornblith, Simon, and Hinton, Geoffrey (2020). “When Does Label Smoothing Help?”.
> 	[7]. Kumar, Aviral, Sarawagi, Sunita, and Jain, Ujjwal (2018). “Trainable Calibration Measures for Neural Networks from Kernel Mean Embeddings.”
> 	[8]. Mukhoti, Jishnu, Kulharia, Viveka, Sanyal, Amartya, Golodetz, Stuart, Torr, Philip H. S., and Dokania, Puneet K. (2020). “Calibrating Deep Neural Networks using Focal Loss.”
>     [9]. McMahan, H. B., Moore, E., Ramage, D., Hampson, S., & Agüera y Arcas, B. (2023). Communication-Efficient Learning of Deep Networks from Decentralized Data.
>     [10]. K. Luo, X. Li, Y. Lan and M. Gao, "GradMA: A Gradient-Memory-based Accelerated Federated Learning with Alleviated Catastrophic Forgetting."
>     [11]. Kim, J., Kim, G., & Han, B. (2022). Multi-Level Branched Regularization for Federated Learning.
>     [12]. McMahan, Brendan, et al. "Communication-efficient learning of deep networks from decentralized data." Artificial intelligence and statistics. PMLR, 2017.
>     [13]. Zhao, Shanshan, Gong, Mingming, Liu, Tongliang, Fu, Huan, and Tao, Dacheng. (2020). “Domain Generalization via Entropy Regularization.”
>     [14]. Schoots, Nandi, and Dylan Cope. (2023). “Low-Entropy Latent Variables Hurt Out-of-Distribution Performance.”
>     [15]. Neo, D., Winkler, S., & Chen, T. (2024). MaxEnt Loss: Constrained Maximum Entropy for Calibration under Out-of-Distribution Shift.

---

### Meta-Review · Area_Chair_RZcK · 2024-12-16

**Metareview:**

The submission worked on a class-prior shift problem in distributed model training environments and proposed a novel and smart solution to better adjust the "test-to-train label density ratio" (in fact, the class label is a discrete RV and thus it is not a density ratio but simply a probability ratio). All the reviewers agreed that it is a good paper and thus should be accepted for publication.

BTW, there is a related work to IW-ERM that you might be interested in: Generalizing importance weighting to a universal solver for distribution shift problems, NeurIPS 2023 (spotlight).

**Additional Comments On Reviewer Discussion:**

The rebuttal addressed (almost all) the concerns from the reviewers.

---

### Decision · Program_Chairs · 2025-01-22

Accept (Poster)